# A Decoupled Learning Framework for Neural Marked Temporal Point Process

## Abstract

The standard neural marked temporal point process employs the Embedding-Encoder-History vector-Decoder (EEHD) architecture, wherein the history vector encapsulates the cumulative effects of past events. However, due to the inherent imbalance in event categories in real-world scenarios, the history vector tends to favor more frequent events, inadvertently overlooking less common yet potentially significant ones, thereby compromising the model's overall performance. To tackle this issue, we introduce a novel decoupled learning framework for neural marked temporal point process, where each event type is modeled independently to capture its unique characteristics, allowing for a more nuanced and equitable treatment of all event types. Each event type boasts its own complete EEHD architecture, featuring scaled-down parameters due to the decoupling of temporal dynamics. This decoupled design enables asynchronous parallel training, and the embeddings can reflect the dependencies between event types. Our versatile framework, accommodating various encoder and decoder architectures, demonstrates state-of-the-art performance across diverse datasets, outperforming benchmarks by a significant margin and increasing training speed by up to 12 times. Additionally, it offers interpretability, revealing which event types have similar influences on a particular event type, fostering a deeper understanding of temporal dynamics.

## 1 Introduction

Event sequence is a ubiquitous data structure in real world, such as user behavior sequences, error logs, purchase transaction records and electronic health records (Mannila et al., 1997; Liu et al., 1998; Zhou et al., 2013; Choi et al., 2016; Liu & Huang, 2023). Regardless of the domain, event sequence provides a unified abstraction for these data, with each event being represented by a tuple, consisting of the event type (aka. mark) and the occurrence time. Modeling such temporal data as a stochastic process, one seeks to predict time and type of the future events based on the history, i.e., previously observed sequential events. For example, a history of purchases of a person may tell when the person will buy a new item, and a short message from a famous social media influencer could affect a critical bull or bear in a stock market. Predicting such a future event is often realized as a chance of the event, i.e., a likelihood of a certain type of event at a specific time.

In the realm of modeling and predicting temporal events, neural marked temporal point processes (MTPPs) have emerged as a powerful tool capable of capturing complex dynamics and dependencies within event sequences (Du et al., 2016; Omi et al., 2019; Shchur et al., 2020; Waghmare et al., 2022; Soen et al., 2021; Zhou & Yu, 2023; Mei & Eisner, 2017; Chen et al., 2018). At the core of these models lies the Embedding-Encoder-History vector-Decoder (EEHD) architecture (Shchur et al., 2021; Bosser & Taieb, 2023), which provides a structured framework for encoding event attributes, summarizing historical contexts, and decoding future event predictions. The history vector, a pivotal component within this architecture, serves as a condensed representation of the cumulative effects of past events, playing a crucial role in shaping the model's predictions. However, a fundamental limitation arises when applying this standard EEHD framework to real-world scenarios, where event categories are inherently unbalanced. In such settings, the history vector tends to exhibit a bias towards more frequently occurring events, inadvertently deemphasizing less common yet potentially significant events. This bias not only undermines the model's ability to capture the full diversity and richness of event dynamics but also compromises its overall predictive performance.

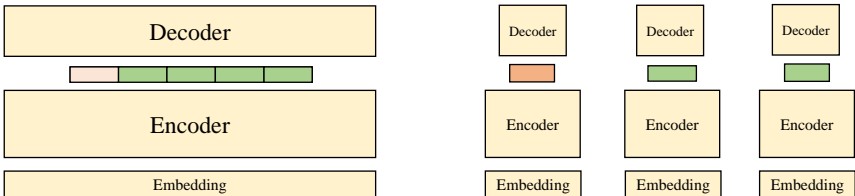

Figure 1: The left figure: the standard learning framework for neural MTPPs, where the history vector tends to overlook those rare events (light orange color) when summarize all past events. The right figure: our proposed decoupled learning framework, where we equip each event type with a complete EEHD architecture and each history vector is dedicated to summarize only the past events that have influences on the corresponding event type, regardless of their frequency. For those less common yet influencing events, they can be captured with high fidelity by our framework (orange color).

To mitigate this challenge and unlock the full potential of neural MTPPs, we introduce a novel decoupled learning framework tailored specifically for this task. Our approach represents a paradigm shift from the traditional monolithic modeling of event sequences to a more nuanced, event-type-specific approach. By modeling each event type individually, independent of the others, we aim to eliminate the bias inherent in the standard EEHD framework and ensure that all event categories, regardless of their frequency, are treated equitably.

Within our decoupled framework, each event type maintains its own dedicated EEHD architecture tailored to its unique characteristics and dynamics, which as a result can have smaller parameter scales compared to that used for full dynamics. The event type embeddings learned in each EEHD can reflect their influences on the corresponding event type, providing valuable insights into the underlying mechanisms driving the temporal point process. Moreover, the decoupled architecture enables asynchronous parallel training for these individual models, significantly enhancing computational efficiency. Our framework can accommodate various encoder and decoder architectures, without imposing rigid constraints. This feature enables researchers and practitioners to experiment with different design choices, optimizing their models for specific tasks and domains.

To summarize, our contributions are as follows:

- We propose a novel decoupled learning framework for neural MTPPs, which revolutionizes the traditional monolithic modeling by disentangling it into event-type-specific individual modeling, thereby enabling an efficient learning for event-type-specific temporal dynamics.

- The decoupled framework allows for asynchronous parallel training and provides interpretability of the dependencies among different event types. Meanwhile, it's general for not imposing restrictions on the encoder and decoder architectures.

- Through extensive experimentation on both real-world and synthetic datasets, we show that our approach attains state-of-the-art performance on standard prediction tasks. Moreover, we showcase a significant enhancement in training speed, achieving up to a 12-fold increase compared to the benchmark model. Also, we demonstrate the interpretability of our framework by analyzing how different event types influence each other.

## 2 METHOD

To ensure that all event types, regardless of their frequency, are adequately represented and considered, we decouple the learning of neural MTPPs from the perspective of event types, with each having its own dedicated EEHD architecture, as shown in Fig. 2. During the decoupling process, we also consider the training efficiency and the interpretability of the dependencies among event types.

***Notation*** Suppose we have $l$ sequential historical events $\{(k_i, t_i)|k_i \in \{1, \cdots, K\}\}_{i=1}^{l}$, and we are aiming at predicting the $(l+1)^{th}$ event, where $K$ is the number of event types, and $k_i, t_i$ are the event type and occurrence time of the $i^{th}$ event, respectively.

## 2.1 EMBEDDING LAYER

***Global embedding*** In standard neural MTPPs, the embedding layer assigns each event type a vector representation and the learned representations can naturally group similar event types according to the spirit of *Word2vec* (Mikolov et al., 2013). However, this embedding style can not reflect the dependencies between different event types. For example, which event types have similar influences on the given event type $k$? We can view the embedding in standard neural MTPPs as *global embedding*, i.e., the vector representations of different event types are in the same vector space.

***Local embedding*** To reflect the relationships among event types, our idea is to create $K$ vector spaces and in each, an event type will have a vector representation, which we call *local embedding*. In this embedding fashion, an event type can have different vector representations in different vector spaces, indicating that it can have different influences on different event types. In vector space of event type $k$, close vector representations indicate that the corresponding event types have similar influences on event type $k$, according to the spirit of *Word2vec*.

Event embedding consists of two parts, the type embedding and the time embedding. Let $z_m^k(\hat{k}) \in \mathbb{R}^{d_m}$ and $z_t^k(\hat{t}) \in \mathbb{R}^{d_t}$ denote the embedding of event type $\hat{k}$ and event time $\hat{t}$ in the vector space of event type $k$ respectively, then we obtain the embedding of the $i^{th}$ event $(k_i, t_i)$ in the vector space of event type $k$ by aggregating the type and time embedding as follows,

$$e^k(i) = z_m^k(k_i) \oplus z_t^k(t_i) \tag{1}$$

where $\oplus$ is usually realized by summation or concatenation. As a comparison, the *global embedding* runs as follows, where only one vector space exists.

$$e(i) = z_m(k_i) \oplus z_t(t_i) \tag{2}$$

The type embedding $z_m(\hat{k})$ is usually realized by looking up the $\hat{k}^{th}$ row of the trainable embedding table $M \in \mathbb{R}^{K \times d_m}$, and the time embedding $z_t(\hat{t})$ is usually realized by sinusoidal functions in literature Zuo et al. (2020); Zhang et al. (2020).

In practice, the dependencies among event types are typically sparse, meaning that for a given event type, only a limited number of event types exert influences on it. Under such a setting, identifying the influencing event types requires only a small dimension $d_m$ in our local embedding, even as low as 1, which greatly improves the learning efficiency.

## 2.2 SEQUENCE ENCODER

***Global encoding*** To evaluate the impacts of historical events $\{(k_i, t_i)\}_{i=1}^l$, the sequence encoder in the standard learning framework encodes all past events into one single history vector $h_l$, which we call *global encoding* and can be summarized as follows:

$$h_l = Encoder(e(1) \cdots, e(i), \cdots, e(l)) \tag{3}$$

where $e(i)$ is the embedding of the $i^{th}$ event $(k_i, t_i)$ and $Encoder$ is usually realized by the recurrent neural network, Transformer or their variants (Bosser & Taieb, 2023). As aforementioned, event types often exhibit a categorical imbalance. This poses a challenge to the standard learning framework, as the history vector, when summarizing event history, tends to give preferential treatment to more common events, to the detriment of those that occur less frequently.

***Local encoding*** To ensure that all event types, irrespective of their frequency, are sufficiently represented and taken into account, we equip each event type with its own dedicated encoder to encode its unique temporal dynamics and dependencies, which we call *local encoding*, i.e.,

$$h_l^k = Encoder^k(e^k(1) \cdots, e^k(i), \cdots, e^k(l)) \tag{4}$$

This design not only enables the model to capture the intricacies of each individual event type but also allows for a more lightweight encoder architecture because $Encoder^k$ can now concentrate solely on the historical events that interest event type $k$.

## 2.3 EVENT DECODER

To model sequences that have $K$ event types (aka. marks), neural MTPP models typically characterize the future dynamics by $K$ conditional intensity functions $\{\lambda_k(t|\mathcal{H}_t)|k \in \{1, \cdots, K\}\}$,

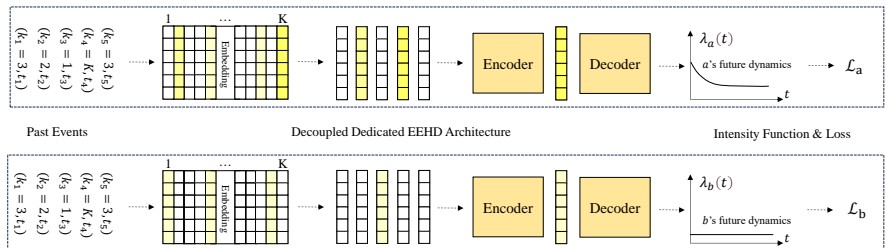

Figure 2: An overview of our proposed decoupled learning framework. Each event type boasts its own complete EEHD architecture (we show that for event type $a$ and $b$). An event type can have different vector representations in different EEHD spaces, indicating that it can have different influences on different event types. For example, the embedding of event type 1 is painted white in $a$'s EEHD space, meaning it has no influence on event type $a$ (brighter color marks stronger influence). Each individual EEHD architecture is to capture the unique characteristics and dynamics of the corresponding event type. Compared with event type $b$, the future dynamism of event type $a$ appears more active (with a greater intensity function) as there are more influencing past events. The overall loss $\mathcal{L}$ is decoupled, which allows for asynchronous parallel training.

where $\mathcal{H}_t$ is the events that occurred before time $t$. $\lambda_k(t|\mathcal{H}_t)$ is defined as the occurrence rate $\mathrm{P}\{\text{event of type } k \text{ occurs in } [t, t+dt)|\mathcal{H}_t\}/dt$, where $[t, t+dt)$ is an infinitesimal time window and P denotes the conditional probability. In literature, there are some other functions employed to describe the future dynamics, e.g., cumulative hazard function (Omi et al., 2019), probability density function (Shchur et al., 2021), etc. All of them can be converted to the intensity function given their mathematical relationships (Bosser & Taieb, 2023). Hence, we only show the case of intensity function in the following.

***Global decoding*** Conditioned on the history vector $\boldsymbol{h_l}$, the decoder is about to decode the $(l+1)^{th}$ event. The decoding process in the standard learning framework can be summarized as follows,

$$\lambda_k(t|\mathcal{H}_t) = \sigma(NN_k(\boldsymbol{h_l}, t)) \tag{5}$$

where $\sigma$ is an activation function to ensure the positive constraint of the intensity function and $NN_k$ is a neural network, e.g., multi-layer perceptron. In the standard learning framework, the history vector $\boldsymbol{h_l}$ is shared across different intensity decoders $\{NN_k\}_{k=1}^K$, and each intensity decoder takes out the information of interests from $\boldsymbol{h_l}$ to generates the corresponding intensity $\lambda_k(t)$. We call this decoding style *global decoding*, whose major challenge is that information in the history vector $\boldsymbol{h_l}$ is mostly dominated by events that occur frequently.

***Local decoding*** In our decoupled learning framework, the information required to decode for each individual event type is ready, i.e., $\{\boldsymbol{h_l^k}\}_{k=1}^K$, yielding the *local decoding* as follows,

$$\lambda_k(t|\mathcal{H}_t) = \sigma(NN_k(\boldsymbol{h_l^k}, t)) \tag{6}$$

## 2.4 ASYNCHRONOUS PARALLEL TRAINING

Suppose we observe a sequence of $L$ events $\mathrm{S} = \{(k_l, t_l)\}_{l=1}^L$ in the time period $[0, T]$, a neural MTPP model is typically trained via the maximum likelihood estimation (MLE), where the log likelihood is calculated as follows,

$$\log \mathcal{L}(\mathrm{S}|\theta_1, \cdots, \theta_K) = \sum_{l=1}^L \log \lambda_{k_l}(t_l|\mathcal{H}_{t_l}) - \int_0^T \sum_{k=1}^K \lambda_k(t|\mathcal{H}_t) dt \tag{7}$$

where $\{\theta_k | k \in \{1, \cdots, K\}\}$ denote the parameters required to train in each individual EEHD architecture. This objective function signifies that the intensity should be large at the occurrence timestamp of each event type (the first term) and small at other timestamps (the second term). Equation 7 also conveys that we should first collect the intensities together from $K$ EEHD architectures, then compute the gradients and distribute the gradients back to update each EEHD model. This is a synchronous training process for multiple models, which will largely diminish the overall training

efficiency. Luckily, this issue can be addressed by the decoupling of the standard objective function, as shown in Equation 8.

$$\log \mathcal{L}(S|\theta_1, \cdots, \theta_K) = \sum_{k=1}^{K} \log \mathcal{L}_k(S|\theta_k), \quad \frac{\partial \log \mathcal{L}(S|\theta_1, \cdots, \theta_K)}{\partial \theta_k} = \frac{\partial \log \mathcal{L}_k(S|\theta_k)}{\partial \theta_k}$$

$$\log \mathcal{L}_k(S|\theta_k) = \sum_{l=1}^{L} \delta(k_l = k) \log \lambda_{k_l}(t_l|\mathcal{H}_{t_l}) - \int_0^T \lambda_k(t|\mathcal{H}_t)dt \tag{8}$$

where $\delta(k_l = k) = 1$ if $k_l = k$ else 0. Equation 8 tells us the synchronization is not really necessary as the gradients for the $k^{th}$ EEHD model can actually be obtained by the component log likelihood $\mathcal{L}_k$, without the need for the total log likelihood $\mathcal{L}$, which allows for asynchronous parallel training and therefore significantly enhances computational efficiency.

## 2.5 DISCUSSION

In this section, we discuss 1) the decoupling of the standard thinning algorithm (Rasmussen, 2018; Mei et al., 2020); 2) the connection of our proposed framework with the Hawkes process (Hawkes, 1971a;b); 3) the connection of our proposed framework with the standard learning framework.

***Sampling algorithm*** Thinning algorithm is widely adopted to draw sequences from point processes. The standard thinning algorithm can be summarized as follows,

$$\lambda(t) = \sum_{k=1}^{K} \lambda_k(t), \quad t \sim Thinning(\lambda(t)), \quad k = \arg\max_{\hat{k}} \frac{\lambda_{\hat{k}}(t)}{\lambda(t)} \tag{9}$$

where we see the sampling of time $t$ is based on the total intensity. One question arises: can we decouple the standard sampling algorithm such that it can be based on the component intensity $\lambda_k(t)$? Algorithm described in Equation 10 answers this.

$$t_k \sim Thinning(\lambda_k(t)), \quad k = \arg\min_{\hat{k}} t_{\hat{k}}, \quad t = t_{\hat{k}} \tag{10}$$

The intuition for the decoupled sampling algorithm in Equation 10 is that the $K$ event types are now in a race to see who generates the next event first. (Typically, the winning type will have relatively high intensity.) Now we can run the thinning algorithm in each individual EEHD model, which can be done in parallel for efficiency. We theoretically show that the sampling algorithms in Equation 9 and 10 are equivalent, see the proof in Appendix.

**Theorem 1.** *The sampling algorithm given by Equation 9 is theoretically equivalent to the decoupled sampling algorithm given by Equation 10.*

***Connection to Hawkes Process*** Hawkes process, a widely studied stochastic process, specifies the conditional intensity function as follows ($\mu_k \geq 0$ is the base intensity),

$$\lambda_k(t|\mathcal{H}_t) = \mu_k + \sum_{t_i < t} \phi_k(k_i, t_i) := \alpha_{k,k_i} \exp(-\beta_{k,k_i}(t - t_i)) \tag{11}$$

We see the impact of historical event $(k_i, t_i)$ on the occurrence of event type $k$ is explicitly characterized by $\alpha_{k,k_i}$ and $\beta_{k,k_i}$, where $\alpha_{k,k_i} \geq 0$ indicates how significantly event type $k_i$ will influence the occurrence of event type $k$, and $\beta_{k,k_i} \geq 0$ shows how this influence decays over time. From the perspective of *local embedding*, the two parameters $[\alpha_{k,k_i}, \beta_{k,k_i}] \in \mathbb{R}^2$ can be considered as the vector representation of event type $k_i$ in the vector space of event type $k$. And $\phi_k(k_i, t_i)$ can be considered as the dedicated encoder for event type $k$ ($Encoder^k$ in Equation 4). Hence, we can view our framework as a generalized neural version of the Hawkes Process.

***Connection to the standard learning framework*** Although the history vector in standard learning framework can be practically biased against event types with different frequencies of occurrence, we show that the standard learning framework is theoretically equivalent to the decoupled one.

**Theorem 2.** *The standard learning framework given by Equation 2, 3 and 5 is theoretically equivalent to the proposed decoupled learning framework characterized by Equation 1, 4 and 6.*

Table 1: Dataset statistics and hyperparameters. The statistics present the number of event types, the count of event sequences, the average sequence length, and the average inter-event times. The hyperparameters are listed for Dec-IFL and IFL (first three) and for Dec-THP and THP (last two).

| Dataset | Statistics | | | | Hyperparameters | | | | | | | | |
|---|---|---|---|---|---|---|---|---|---|---|---|---|---|
| | #Types | #Seqs | #Avg. E | Avg. T | $d_m$ | | $d_h$ | | $N_1$ | | $d$ | | $N_2$ |
| SOflow | 22 | 6633 | 72 | 12.84 | 1 | 16 | 4 | 32 | 1 | 1 | 8 | 128 | 3 | 4 |
| MIMIC | 75 | 715 | 4 | 0.29 | 1 | 32 | 4 | 32 | 1 | 1 | 4 | 128 | 3 | 4 |
| MOOC | 97 | 7047 | 56 | 4.32 | 1 | 32 | 4 | 64 | 1 | 2 | 8 | 256 | 3 | 5 |
| ICEWS | 201 | 1352 | 38 | 0.58 | 1 | 32 | 4 | 64 | 1 | 2 | 8 | 256 | 3 | 6 |

*Proof.* We first show that our framework is no weaker in expressive power than the standard learning framework. Given an input, in order to have the same output as the standard learning framework, we only need to let $z_m^k, z_t^k$ in Equation 1 equal to $z_m, z_t$ in Equation 2, $Encoder^k$ in Equation 4 equal to $Encoder$ in Equation 3, and $NN_k$ in Equation 6 equal to that in Equation 5.

Next, we show that the standard learning framework is no weaker in expressive power than our framework. Firstly, we let $NN_k$ in Equation 5 equal to that in Equation 6. Now we only need to show that the history vector $h$ in the standard learning framework is not weaker in expressive power than those $K$ history vectors $\{h^k | k \in \{1, \cdots, K\}\}$ in our framework. Let $z_m, z_t$ in Equation 2 equal to $z_m^1 || \cdots || z_m^K, z_t^1 || \cdots || z_t^K$ in Equation 1, where $||$ is the concatenation operation. Then according to the universal approximation theory of the widely used sequence encoders such as RNN, Transformer and state space model (Schäfer & Zimmermann, 2006; Gu et al., 2020; Furuya et al., 2024), the $Encoder$ in Equation 3 can produce the output $h = h^1 || \cdots || h^K$. □

In fact, the grouped decoupling can unify these two frameworks. Specifically, grouped decoupling divides all event types into several groups, e.g., according to the occurrence frequency, and build one EEHD model for each group. If we set the number of groups to 1, then it is the standard learning framework. And if we set the number of groups to $K$, then it is our proposed framework.

## 3 EXPERIMENTS

### 3.1 EXPERIMENTAL SETUP

***Datasets*** We adopt four real-world datasets for evaluation. SOflow (Bosser & Taieb, 2023) is records of the time when users received a specific badge on the question-answering website Stack Overflow and the event type is the badge received. MIMIC medical dataset (Bosser & Taieb, 2023) collects patients' visit to a hospital's ICU in a seven-year period. Each patient's visits constitute an event sequence, with each visit event containing a timestamp and a diagnosis. MOOC (Bosser & Taieb, 2023) is records of student's actions on an online course system and each action is associated with a timestamp. ICEWS (Boschee et al., 2015) consists of interactions between socio-political actors (i.e., cooperative or hostile actions between individuals, groups, sectors and nation states) in year 2018, where the events are extracted from news articles by the Integrated Crisis Early Warning System. Dataset statistics are shown in Table 1. Each dataset is split into training/validation/testing set according the number of event sequences, with each part accounting for 60%/20%/20%, respectively.

***Baselines*** To demonstrate the efficacy of our proposed decoupled learning framework, we decouple two state-of-the-art models: IFL (Shchur et al., 2020) and THP (Zuo et al., 2020), yielding Dec-IFL and Dec-THP. IFL employs GRU (Chung et al., 2014)) as the encoder and formulates the temporal dynamics via a density function based on normalizing flows (Rezende & Mohamed, 2015), the mixture log-normal distribution. THP uses Transformer to encode the historical impacts and characterizes the temporal dynamics by a monotonic intensity function. Beyond IFL and THP, we also report the performance of RMTPP (Du et al., 2016), NHP (Mei & Eisner, 2017), SAHP (Zhang et al., 2020) and ODETPP (Song et al., 2024). RMTPP models a monotonic intensity function by recurrent neural network. NHP designs a continuous-time LSTM to learn time-evolving history vectors, while ODETPP uses the neural ODE. SAHP also uses Transformer as the encoder like THP, but learns a bounded monotonic intensity function to limit the range of the intensity.

Table 2: Comparison of the performance on event prediction, evaluated by negative log likelihood (NLL), weighted F1 score (F1) and mean square error (MSE). For NLL and MSE, the lower the better. For F1, the higher the better. The results are averaged by 10 runs with different random seeds. Bold number indicates that the model is better than its counterpart.

| Model | SOflow | | | MIMIC | | | MOOC | | | ICEWS | | |
|---|---|---|---|---|---|---|---|---|---|---|---|---|
| | NLL | F1 | MSE | NLL | F1 | MSE | NLL | F1 | MSE | NLL | F1 | MSE |
| RMTPP | 246.0 | 30.1 | 4.10 | 7.3 | 64.4 | 0.44 | 226.0 | 38.5 | 6.94 | -144.8 | 26.4 | 0.98 |
| NHP | 238.2 | 30.8 | 3.59 | 7.1 | 64.8 | 0.38 | 210.3 | 39.0 | 5.73 | -180.5 | 27.2 | 0.73 |
| SAHP | 233.4 | 31.6 | 3.76 | 6.9 | 65.3 | 0.37 | 199.4 | 39.9 | 5.67 | -185.4 | 28.9 | 0.89 |
| ODETPP | 231.7 | 31.0 | 3.12 | 6.6 | 64.9 | 0.34 | 193.8 | 38.7 | 5.16 | -193.4 | 27.5 | 0.77 |
| IFL | 225.3 | 30.7 | 2.25 | 6.0 | **65.9** | 0.28 | 185.7 | 39.2 | 4.80 | -215.7 | 27.9 | 0.67 |
| THP | 235.4 | 31.5 | 3.55 | **6.8** | 65.7 | **0.35** | 202.3 | 39.6 | 6.11 | -190.0 | 29.4 | 0.84 |
| Dec-IFL | **219.3** | **32.2** | **2.05** | 6.0 | 65.6 | **0.27** | **181.1** | **40.5** | **4.12** | **-253.6** | **29.1** | **0.51** |
| Dec-THP | **225.7** | **32.4** | **2.97** | 6.9 | 65.7 | 0.37 | **187.4** | **41.3** | **5.23** | **-209.3** | **30.6** | **0.70** |

Table 3: Type-specific prediction accuracy comparison on dataset SOflow, which has a total of 22 event types. We list their frequency and rank them in ascending order, shown in rows 1 and 5.

| | 7 | 17 | 59 | 107 | 189 | 251 | 333 | 404 | 503 | 740 | 879 |
|---|---|---|---|---|---|---|---|---|---|---|---|
| IFL | 0.00 | 0.01 | 0.00 | 0.00 | 0.00 | 0.00 | 0.00 | **0.30** | **0.07** | 0.00 | 0.00 |
| Dec-IFL | 0.00 | **0.19** | 0.00 | 0.00 | 0.00 | **0.06** | 0.00 | 0.21 | 0.05 | 0.00 | 0.00 |
| THP | 0.00 | 0.10 | 0.00 | **0.14** | 0.00 | 0.00 | **0.09** | 0.18 | 0.10 | 0.00 | **0.16** |
| Dec-THP | 0.00 | **0.15** | 0.00 | 0.00 | 0.00 | 0.00 | 0.00 | **0.37** | **0.13** | **0.08** | 0.13 |
| | 1358 | 1771 | 1791 | 2050 | 2085 | 2529 | 5209 | 5537 | 6477 | 21024 | 39335 |
| IFL | 0.00 | **0.02** | 0.06 | 0.18 | 0.04 | 0.04 | 0.00 | 0.04 | 0.24 | 0.17 | **0.37** |
| Dec-IFL | 0.16 | 0.00 | **0.19** | **0.23** | **0.14** | **0.08** | 0.00 | **0.12** | **0.33** | **0.23** | 0.34 |
| THP | 0.06 | **0.07** | 0.10 | 0.10 | 0.07 | 0.06 | **0.11** | 0.10 | 0.22 | 0.19 | 0.37 |
| Dec-THP | **0.10** | 0.03 | **0.15** | **0.22** | **0.12** | **0.07** | 0.04 | **0.14** | **0.30** | **0.21** | 0.37 |

When applying our framework, we set the same hyperparameters for $K$ decoupled EEHD models and they are trained asynchronously. Specifically, when training for Dec-IFL, we mainly tune the dimension of the type embedding $d_m$, the dimension of the history vector $d_h$ and the number of layers $N_1$ stacked in the GRU architecture. For transformer-based Dec-THP, the dimensions of the type embedding and history vector are set to the same value $d$. We mainly tune the dimension $d$ and the number of blocks $N_2$ stacked in the Transformer architecture. These hyperparameters are reported in Table 1, where we see that the $K$ decoupled EEHD models have much smaller parameter scales compared to the traditional one due to the decoupled dynamics for each event type. For other baselines, we follow the hyperparameter settings in their paper.

## 3.2 PREDICTION RESULTS

The predictive capability of a sequence model can be assessed by its ability to forecast the subsequent event based on the historical sequence of events. We use the weighted-F1 score to evaluate the accuracy of event type prediction and use mean square error to evaluate the error of event time prediction. Besides, we use negative log likelihood (NLL) to evaluate the event distribution prediction, which simultaneously considers the prediction of event type and occurrence time. The results on the four real-world datasets are summarized in Table 2. We see Dec-IFL and Dec-THP outperform their counterparts IFL and THP significantly in most datasets. However, it appears that the decoupled learning framework does not yield effective results on the MIMIC dataset. This is attributed to the exceptionally short average sequence length of 4 in MIMIC, which negates the frequency bias of the history vector within the standard learning framework. To delve deeper into the issue of frequency bias, Table 3 presents a detailed analysis of prediction accuracy for each event type on the SOflow dataset. Here, Dec-IFL outperforms IFL for 10 event types, whereas IFL surpasses Dec-IFL for only 4 event types. Similar conclusions can be drawn when comparing Dec-THP with THP. To see the overall and type-specific performance of our framework on more datasets, readers can refer to Appendix B.

$$[0. , 0. , 1.5, 0. , 0. ] \quad [0. , 0. , 1.5, 1.5, 0. ] \quad [0. , 1.5, 0. , 0. , 0. ] \quad [0., 1.5, 1.5, 1.5, 0.]$$
$$[0. , 0. , 1.5, 0. , 0. ] \quad [0. , 1.5, 0. , 0. , 0. ] \quad [0. , 1.5, 0. , 0., 0. ] \quad [0., 1.5, 0., 1.5, 0.]$$
$$[0. , 0. , 1.5, 0. , 0. ] \quad [0. , 1.5, 0. , 0. , 0. ] \quad [0. , 1.5, 0. , 0., 0. ] \quad [0., 1.5, 1.5, 1.5, 0.]$$
$$[0. , 0. , 1.5, 0. , 0. ] \quad [0. , 1.5, 0. , 0. , 0. ] \quad [0. , 1.5, 0. , 0., 0. ] \quad [0., 1.5, 1.5, 0., 0.]$$
$$[0. , 0. , 1.5, 0. , 0. ] \quad [0. , 0. , 1.5, 1.5, 0. ] \quad [0. , 1.5, 1.5,1.5, 0. ] \quad [0., 1.5, 0., 1.5, 0.]$$

$$[ 0.1, 0.0, -0.2, 0.1, 0.1] \quad [-0.1, -0.3, 0.2, 0.2, -0.1] \quad [ 0.1, -0.3, 0.6, 0.8, 0.2] \quad [ 0.2, -0.4, -0.8, -0.8, 0.3]$$
$$[ 0.0, 0.1, 0.2, -0.1, 0.1] \quad [ 0.2, -0.2, -0. , -0., 0.3] \quad [-0.7, 0.5, -0.3, -0.2, -0.6] \quad [-0.3, 0.4, -0.6, 1.1, -0.2]$$
$$[ 0.1, 0.1, -0.2, 0.1, 0.2] \quad [ 0.3, -0.3, -0.1, 0., 0.3] \quad [-0.4, 0.4, -0.2, -0.2, -0.5] \quad [ 0.8, -0.5, -0.5, -0.6, 0.7]$$
$$[ 0.2, 0.0, -0.3, 0.1, 0.1] \quad [ 0.2, -0.3, 0.1, -0., 0.2] \quad [-0.7, 0.6, -0.2, -0.3, -0.7] \quad [ 0.6, -0.7, -1.0, 0.4, 0.5]$$
$$[ 0.1, 0.1, -0.1, 0.3, 0.1] \quad [-0.2, -0.1, 0.2, 0.3, -0.3] \quad [-0., 0.3, 0.6, 0.8, -0.1] \quad [ 0.4, -0.6, 0.3, -0.9, 0.3]$$

Figure 3: The four matrices displayed above present the ground truth influences among event types within datasets Haw1, Haw2, Haw3, and Haw4, where the element in the $i^{th}$ row and $j^{th}$ column indicates how significantly event type $j$ influences event type $i$. The four matrices displayed below are the embeddings learned by Dec-IFL on dataset Haw1, Haw2, Haw3, and Haw4, where the element in the $i^{th}$ row and $j^{th}$ is the embedding of event type $j$ in the vector space of event type $i$. (The embedding dimension is 1 and the embeddings are rounded to one decimal place.)

Table 4: Comparison of training parameter count and training speed per epoch (in seconds).

| Datasets | #Parameter | | Ratio | Speed | | Ratio | #Parameters | | Ratio | Speed | | Ratio |
|---|---|---|---|---|---|---|---|---|---|---|---|---|
| | IFL | Dec-IFL | | IFL | Dec-IFL | | THP | Dec-THP | | THP | Dec-THP | |
| SOflow | 145K | 1K | 145 | 8 | 1 | 8 | 0.8M | 6K | 133 | 24 | 4 | 6 |
| MIMIC | 0.5M | 1K | 500 | 0.3 | 0.15 | 2 | 0.8M | 1K | 800 | 0.5 | 0.2 | 2.5 |
| MOOC | 1.2M | 1K | 1200 | 9 | 1 | 9 | 4.0M | 6K | 666 | 27 | 4 | 6.8 |
| ICEWS | 2.6M | 1K | 2600 | 6 | 0.5 | 12 | 4.8M | 6K | 800 | 18 | 2 | 9 |

## 3.3 INTERPRETABILITY AND TRAINING EFFICIENCY

To better evaluate the interpretabiity of our proposed framework, we utilize four synthetic datasets generated by the Hawkes Process (Eq. 11), specifically named Haw1, Haw2, Haw3, and Haw4, where the ground truth influences among event types are known. In Fig. 3, the upper four matrices present the configurations of the parameters $\alpha_{i,j}$ for the four Hawkes datasets. The parameters $\beta_{i,j}$ are uniformly set to 2.5 across all instances. The parameter $\alpha_{i,j}$ quantifies the magnitude of influence that event type $j$ exerts on event type $i$, analogous to the embedding of event type $j$ in the vector space of event type $i$. We extract the embeddings learned by Dec-IFL on these four Hawkes datasets and present them by the lower four matrices in Fig. 3. Our findings indicate that these learned embeddings accurately capture the underlying dependencies among event types. For instance, in dataset Haw4, event types 2, 3, and 4 all exert influences on event type 1, which is reflected in the proximity of their embeddings in the vector space of event type 1. This observation aligns with the spirits underlying *Word2vec*: event types that exert similar influences on a given event type $i$ tend to have embeddings that are close in the vector space of event type $i$. For more demonstrations of interpretability, readers can refer to Appendix C.

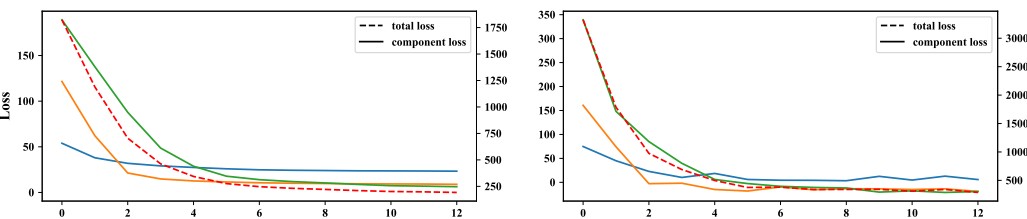

Figure 4: The total loss and component loss at each epoch for Dec-IFL (left figure) and IFL (right figure) on dataset SOflow. For each figure, see y-axis on the left for component loss and y-axis on the right for total loss. There are a total of 22 event types in dataset SOflow, we only selectively plotted the component losses for three event types for clarity.

The proposed decoupled learning framework allows for asynchronous parallel training, which greatly enhances the training efficiency. Table 4 presents the number of parameters required for learning and the training speed. Note that the reported parameter counts for Dec-IFL and Dec-THP pertain to a single decoupled EEHD model. And the reported training speed is the lowest training speed for $K$ decoupled EEHD models. As evident from Table 4, a single decoupled EEHD model boasts a substantially smaller parameter scale compared to traditional models, owing to the decoupled dynamics for each event type, which drastically improves the training speed. Another advantage of asynchronous parallel training is its capacity to ensure full convergence of each individual decoupled EEHD model with respect to component loss $-\log \mathcal{L}_k(\mathrm{S})$, as illustrated in the left figure in Fig. 4. Conversely, in traditional models, while the overall loss $-\log \mathcal{L}(\mathrm{S})$ may indicate convergence, the component loss for each event type may not have achieved convergence (the right figure of Fig. 4).

## 4 RELATED WORK

Depending on whether the EEHD architecture is applied or not, neural MTPP models can be categorised into two groups. Most models fall into the category using the EEHD architecture, while there are also some models that do not, such as models that use graph structures to represent event types and their relationships (Trivedi et al., 2019; Zhang & Yan, 2021; Dash et al., 2022), and models that use case-based reasoning (Liu, 2024a). Among the EEHD-based models, researchers have worked on advancing the encoder and decoder to capture more complex dynamics. Popular encoders include recurrent neural networks (Du et al., 2016; Omi et al., 2019; Shchur et al., 2020), attention based networks (Zuo et al., 2020; Zhang et al., 2020; Gu, 2021; Shou et al., 2023), and state space model based networks (Gao et al., 2024). These encoders generate the history vector that remains static until the occurrence of the subsequent event. In contrast, continuous-time encoders, exemplified by continuous-LSTM (Mei & Eisner, 2017) and neural ODEs (Chen et al., 2018; 2021; Song et al., 2024), allow the history vector to evolve over time. Regarding decoders, most neural MTPP models rely on the intensity function (Du et al., 2016; Mei & Eisner, 2017; Zhang et al., 2020; Soen et al., 2021; Ding et al., 2023), which is formulated through a predefined parameterized function (e.g., a neural network) and the history vector is used to decode the function parameters. There are also some intensity-free decoders designed for closed-form computing of the likelihood (Omi et al., 2019; Shchur et al., 2020; Waghmare et al., 2022; Liu et al., 2023; Liu, 2024b). For example, FullyNN (Omi et al., 2019) models the cumulative hazard function by a monotonic neural network, where the neural weights are all constrained to be positive; IFL (Shchur et al., 2020) models the probability density function by a log-normal mixture distribution. Based on their mathematical relationships, the intensity models and intensity-free models can be converted interchangeably (Lin et al., 2021).

To our knowledge, we are pioneers in addressing the imbalance of event categories, prompting us to decouple neural MTPPs from the perspective of event types. However, decoupling can have many aspects. For example, some models calculate the cumulative impacts of past events as the sum of the impacts of each individual past events to identify the important past events as well as improve the training efficiency (Liu et al., 2018; Salehi et al., 2019; Zhou & Yu, 2023; Song et al., 2024). This is also a form of decoupling, but for a different purpose than ours. Notably, it compromises model performance as it does not take into account the temporal dependencies between historical events.

## 5 CONCLUSION

In this paper, we have presented a novel decoupled learning framework for neural marked temporal point processes that effectively mitigates the issue of frequency bias inherent in standard approaches. By modeling each event type separately within a complete EEHD architecture, our approach ensures that all event types, regardless of their frequency, are adequately represented and considered during training. The decoupling of the standard monolithic modeling not only enables asynchronous parallel training, significantly improving the training speed, but also allows the embeddings to capture the intricate dependencies between event types, which we believe will have far-reaching practical implications around knowledge discovery in many domains. Our extensive experiments across diverse datasets have conclusively demonstrated that this decoupled learning framework achieves state-of-the-art performance on standard prediction tasks, underscoring its effectiveness. Importantly, the framework's design does not impose rigid constraints on the encoder and decoder architectures, allowing for future improvements and extensions tailored to specific applications.

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

Table A1: Dataset statistics and hyperparameters. For left to right column in statistics: number of event types, number of event sequences, average number of events in one event sequence, and the average inter-event time interval. For left to right column in hyperparameters: the dimension of the type embedding $d_m$, the dimension of the history vector $d_h$, the number of layers $N_1$ stacked in the GRU architecture for Dec-IFL and IFL; the unified vector dimension $d$, the number of blocks $N_2$ stacked in the Transformer architecture for Dec-THP and THP.

| Dataset | Statistics | | | | Hyperparameters | | | | | | | |
|---|---|---|---|---|---|---|---|---|---|---|---|---|
| | #Types | #Seqs | #Avg. E | Avg. T | $d_m$ | | $d_h$ | | $N_1$ | | $d$ | | $N_2$ |
| Haw1 | 5 | 1918 | 12 | 0.27 | 1 | 16 | 4 | 32 | 1 | 1 | 4 | 32 | 3 | 4 |
| Haw2 | 5 | 1035 | 10 | 0.25 | 1 | 16 | 4 | 32 | 1 | 1 | 4 | 32 | 3 | 4 |
| Haw3 | 5 | 1184 | 8 | 0.45 | 1 | 16 | 4 | 32 | 1 | 1 | 4 | 32 | 3 | 4 |
| Haw4 | 5 | 1468 | 8 | 0.16 | 1 | 16 | 4 | 32 | 1 | 1 | 4 | 32 | 3 | 4 |

# Appendix

## A  PROOFS

*Proof.* (of Theorem 1). Recall that in the standard thinning algorithm (Rasmussen, 2018), we sample the time $t$ using an exponential distribution with parameter $\bar{\lambda}$, where $\bar{\lambda}$ is an upper bound of the total intensity $\lambda(t)$. In our decoupled sampling algorithm, the thinning algorithm is running in each individual EEHD model with the component intensity $\lambda_k(t)$. Let $T_k$ be a random variable that follows the exponential distribution with parameter $\tilde{\lambda}_k$, where $\tilde{\lambda}_k$ is an upper bound of the component intensity $\lambda_k(t)$. We next show that $T = \min(T_1, \cdots, T_K)$ still follows the exponential distribution and its parameter $\tilde{\lambda}$ is an upper bound of the total intensity $\lambda(t)$, which then completes the proof.

Since $\{T_k\}_{k=1}^K$ are independent random variables, the distribution function of $T$ can be calculated as follows.

$$
\begin{aligned}
F(T \leq t) &= \mathrm{P}\{\min(T_1, \cdots, T_K) \leq t\} \\
&= 1 - \mathrm{P}\{T_1 > t, \cdots, T_K > t\} \\
&= 1 - \mathrm{P}\{T_1 > t\} \times \cdots \times \mathrm{P}\{T_K > t\} \\
&= 1 - (1 - \mathrm{P}\{T_1 \leq t\}) \times \cdots \times (1 - \mathrm{P}\{T_K \leq t\}) \\
&= 1 - \exp(-\tilde{\lambda}_1 t) \times \cdots \times \exp(-\tilde{\lambda}_K t) \\
&= 1 - \exp(-\sum_{k=1}^K \tilde{\lambda}_k t)
\end{aligned}
\tag{1}
$$

where we see $T$ still follows the exponential distribution and its parameter $\tilde{\lambda}$ satisfies that $\tilde{\lambda} = \sum_{k=1}^K \tilde{\lambda}_k \geq \sum_{k=1}^K \lambda_k(t) = \lambda(t)$. That is, $\tilde{\lambda}$ is an upper bound of the total intensity $\lambda(t)$.

$\square$

## B  ADDITIONAL PREDICTION RESULTS

In the main text of this paper, we only apply our framework to IFL and THP. Here, we further apply our framework to the remaining baselines as a complement to Table 2 in the main text. Please note that our framework is specifically tailored for standard EEHD models, which means the baseline ODETPP (Song et al., 2024) is not included in the list of potential applications due to its non-conformity with the standard EEHD model. Like ours but for a different purpose, ODETPP is also a generalized Hawkes process and therefore does not conform to the standard EEHD model. Using neural ODE (Chen et al., 2018), ODETPP mainly studies how individual historical events influence the overall dynamics. The prediction results of the decoupled models and their counterparts are all summarized in Table A3.

We additionally utilize four synthetic datasets generated by the Hawkes Process (Eq. 11), specifically named Haw1, Haw2, Haw3, and Haw4, for evaluation. The upper four matrices in Fig. 3 present the

Table A2: Predictive performance of Dec-IFL, Dec-THP and baselines on four synthetic datasets Haw1, Haw2, Haw3 and Haw4.

| Model | Haw1 | | | Haw2 | | | Haw3 | | | Haw4 | | |
|---|---|---|---|---|---|---|---|---|---|---|---|---|
| | NLL | ACC | MSE | NLL | ACC | MSE | NLL | ACC | MSE | NLL | ACC | MSE |
| RMTPP | 7.9 | 22.5 | 0.37 | 11.6 | 23.9 | 0.32 | 12.7 | 32.7 | 0.40 | 12.0 | 27.1 | 0.33 |
| NHP | 7.0 | 23.0 | 0.25 | 10.6 | 24.7 | 0.27 | 12.1 | 32.6 | 0.35 | 11.5 | 27.5 | 0.20 |
| SAHP | 7.2 | 24.4 | 0.27 | 11.0 | 25.4 | 0.31 | 12.2 | 34.5 | 0.37 | 11.8 | 27.4 | 0.29 |
| ODETPP | 7.2 | 22.8 | 0.17 | 10.3 | 24.8 | 0.25 | 11.3 | 33.8 | 0.24 | 10.8 | 27.9 | 0.14 |
| IFL | 6.6 | 23.7 | 0.21 | 9.8 | 25.0 | 0.23 | 11.0 | 33.5 | 0.21 | 10.3 | 28.4 | 0.12 |
| THP | 7.3 | 24.3 | 0.31 | 10.8 | 25.3 | 0.28 | 11.8 | 34.4 | 0.31 | 11.1 | 29.3 | 0.28 |
| Dec-IFL | 6.2 | 24.2 | 0.15 | 9.1 | 26.1 | 0.18 | 10.5 | 34.6 | 0.17 | 10.0 | 29.7 | 0.10 |
| Dec-THP | 6.5 | 24.7 | 0.19 | 9.9 | 26.3 | 0.25 | 11.3 | 35.2 | 0.29 | 10.8 | 30.3 | 0.27 |

Table A3: Comparison of the performance on event prediction, evaluated by negative log likelihood (NLL), weighted F1 score (F1) and mean square error (MSE). For NLL and MSE, the lower the better. For F1, the higher the better. The results are averaged by 10 runs with different random seeds and the standard deviations are reported in brackets.

| Model | SOflow | | | MIMIC | | | MOOC | | | ICEWS | | |
|---|---|---|---|---|---|---|---|---|---|---|---|---|
| | NLL | F1 | MSE | NLL | F1 | MSE | NLL | F1 | MSE | NLL | F1 | MSE |
| RMTPP | $246.0_{\pm2.3}$ | $30.1_{\pm0.2}$ | $4.10_{\pm0.3}$ | $7.3_{\pm0.1}$ | $64.4_{\pm0.3}$ | $0.44_{\pm0.03}$ | $226.0_{\pm1.8}$ | $38.5_{\pm0.3}$ | $6.94_{\pm0.2}$ | $-144.8_{\pm4.7}$ | $26.4_{\pm0.2}$ | $0.98_{\pm0.06}$ |
| NHP | $238.2_{\pm2.5}$ | $30.8_{\pm0.2}$ | $3.59_{\pm0.2}$ | $7.1_{\pm0.1}$ | $64.8_{\pm0.4}$ | $0.38_{\pm0.02}$ | $210.3_{\pm2.5}$ | $39.0_{\pm0.4}$ | $5.73_{\pm0.2}$ | $-180.5_{\pm3.6}$ | $27.2_{\pm0.3}$ | $0.73_{\pm0.04}$ |
| SAHP | $233.4_{\pm2.8}$ | $31.6_{\pm0.3}$ | $3.76_{\pm0.3}$ | $6.9_{\pm0.2}$ | $65.3_{\pm0.4}$ | $0.37_{\pm0.03}$ | $199.4_{\pm2.1}$ | $39.9_{\pm0.3}$ | $5.67_{\pm0.3}$ | $-185.4_{\pm4.3}$ | $28.9_{\pm0.2}$ | $0.89_{\pm0.06}$ |
| ODETPP | $231.7_{\pm2.3}$ | $31.0_{\pm0.3}$ | $3.12_{\pm0.2}$ | $6.6_{\pm0.1}$ | $64.9_{\pm0.4}$ | $0.34_{\pm0.02}$ | $193.8_{\pm2.0}$ | $38.7_{\pm0.3}$ | $5.16_{\pm0.2}$ | $-193.4_{\pm4.2}$ | $27.5_{\pm0.4}$ | $0.77_{\pm0.03}$ |
| IFL | $225.3_{\pm1.8}$ | $30.7_{\pm0.3}$ | $2.25_{\pm0.1}$ | $6.0_{\pm0.1}$ | $65.9_{\pm0.3}$ | $0.28_{\pm0.02}$ | $185.7_{\pm1.0}$ | $39.2_{\pm0.3}$ | $4.80_{\pm0.1}$ | $-215.7_{\pm3.3}$ | $27.9_{\pm0.4}$ | $0.67_{\pm0.03}$ |
| THP | $235.4_{\pm3.2}$ | $31.5_{\pm0.4}$ | $3.55_{\pm0.3}$ | $6.8_{\pm0.2}$ | $65.7_{\pm0.4}$ | $0.35_{\pm0.01}$ | $202.3_{\pm2.5}$ | $39.6_{\pm0.4}$ | $6.11_{\pm0.2}$ | $-190.0_{\pm3.2}$ | $29.4_{\pm0.4}$ | $0.84_{\pm0.05}$ |
| Dec-RMTPP | $237.3_{\pm2.1}$ | $30.8_{\pm0.3}$ | $3.55_{\pm0.2}$ | $7.2_{\pm0.1}$ | $64.6_{\pm0.3}$ | $0.40_{\pm0.02}$ | $211.1_{\pm2.0}$ | $39.6_{\pm0.4}$ | $6.12_{\pm0.2}$ | $-163.6_{\pm4.3}$ | $27.3_{\pm0.3}$ | $0.71_{\pm0.04}$ |
| Dec-NHP | $232.7_{\pm2.6}$ | $31.5_{\pm0.4}$ | $3.09_{\pm0.2}$ | $7.1_{\pm0.1}$ | $64.5_{\pm0.2}$ | $0.37_{\pm0.03}$ | $199.5_{\pm2.0}$ | $39.9_{\pm0.3}$ | $5.58_{\pm0.2}$ | $-191.4_{\pm3.5}$ | $28.5_{\pm0.4}$ | $0.62_{\pm0.04}$ |
| Dec-SAHP | $226.4_{\pm3.0}$ | $32.6_{\pm0.4}$ | $3.11_{\pm0.2}$ | $6.8_{\pm0.1}$ | $65.5_{\pm0.3}$ | $0.37_{\pm0.01}$ | $187.7_{\pm1.9}$ | $40.8_{\pm0.3}$ | $5.03_{\pm0.2}$ | $-202.1_{\pm4.0}$ | $30.2_{\pm0.3}$ | $0.75_{\pm0.03}$ |
| Dec-IFL | $219.3_{\pm1.6}$ | $32.2_{\pm0.3}$ | $2.05_{\pm0.1}$ | $6.0_{\pm0.1}$ | $65.6_{\pm0.2}$ | $0.27_{\pm0.02}$ | $181.1_{\pm0.8}$ | $40.5_{\pm0.2}$ | $4.12_{\pm0.1}$ | $-253.6_{\pm4.1}$ | $29.1_{\pm0.5}$ | $0.51_{\pm0.02}$ |
| Dec-THP | $225.7_{\pm2.6}$ | $32.4_{\pm0.5}$ | $2.97_{\pm0.2}$ | $6.9_{\pm0.2}$ | $65.7_{\pm0.3}$ | $0.37_{\pm0.03}$ | $187.4_{\pm2.1}$ | $41.3_{\pm0.4}$ | $5.23_{\pm0.1}$ | $-209.3_{\pm4.5}$ | $30.6_{\pm0.5}$ | $0.70_{\pm0.03}$ |

configurations of the parameters $\alpha_{i,j}$ for the four Hawkes datasets. The parameters $\beta_{i,j}$ are uniformly set to 2.5 across all instances. Table A1 summarizes their statistics and our used hyperparameters. Table A2 reports the results on these four synthetic datasets with respect to the event distribution prediction, event type prediction and event time prediction. For type-specific event type prediction results on datasets MOOC and ICEWS, we show in Figure A1, A2, A3 and A4. We see our framework outperforms its counterparts consistently and significantly.

## C    ADDITIONAL INTERPRETABILITY DEMONSTRATION

It is worth noting that the embeddings in different EEHD models are asynchronously trained, which means that if there are a large number of event types, we can only learn the embedding we are interested, reducing computational costs. We here illustrate the embeddings in some interested vector spaces learned on the socio-political dataset ICEWS. Specifically, we are interested in two event types "Express intent to cooperate" and "Mobilize or increase armed forces". We show the embeddings in the vector space of event type "Express intent to cooperate" in Fig. A6, the embeddings in the vector space of event type "Mobilize or increase armed forces" in Fig. A7, where we have marked some event types that have close embeddings in green. The clustering results show interesting patterns

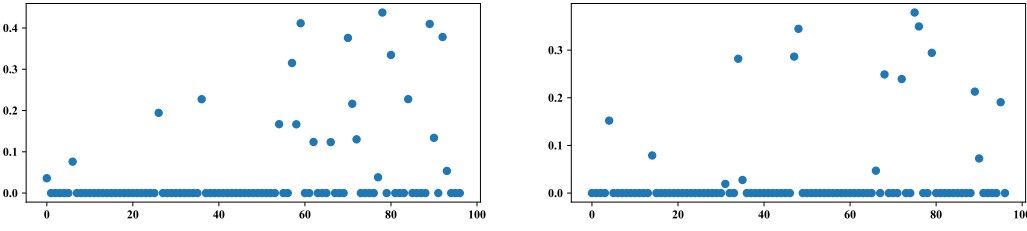

Figure A1: The type-specific event type prediction performance on dataset MOOC. The left figure comes from Dec-IFL and the right figure comes from IFL. The x-axis is the event type, arranged in ascending frequency order.

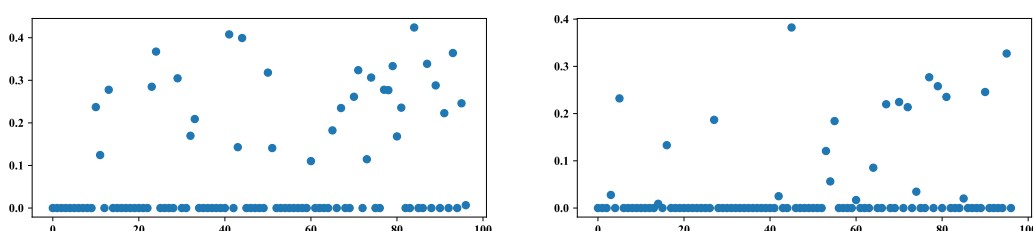

Figure A2: The type-specific event type prediction performance on dataset MOOC. The left figure comes from Dec-THP and the right figure comes from THP. The x-axis is the event type, arranged in ascending frequency order.

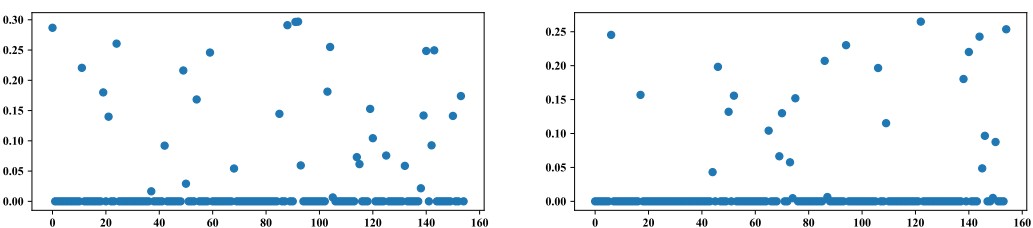

Figure A3: The type-specific event type prediction performance on dataset ICEWS. The left figure comes from Dec-IFL and the right figure comes from IFL. Note some event types never happen in testing sequences. The x-axis is the event type, arranged in ascending frequency order.

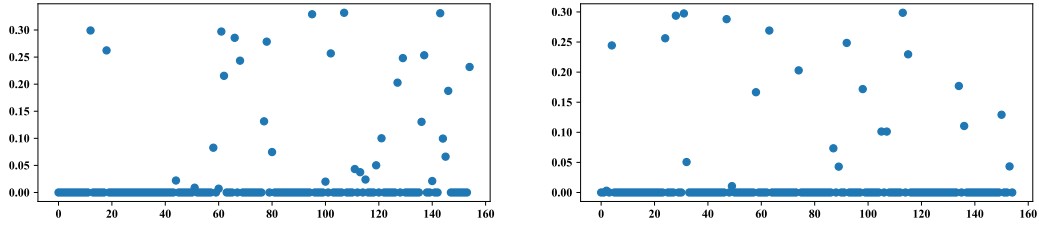

Figure A4: The type-specific event type prediction performance on dataset ICEWS. The left figure comes from Dec-THP and the right figure comes from THP. Note some event types never happen in testing sequences. The x-axis is the event type, arranged in ascending frequency order.

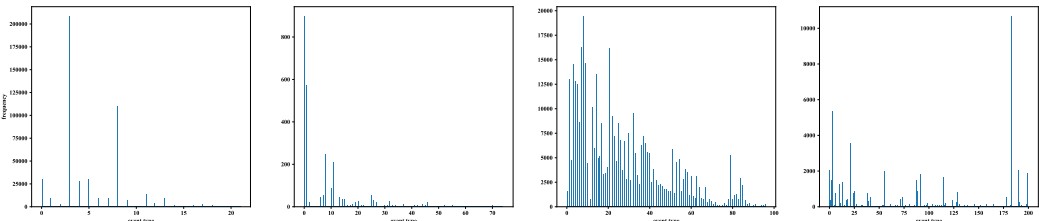

Figure A5: The occurrence frequency of event types over datasets SOflow, MIMIC, MOOC and ICEWS (from left to right). The entropy of these four frequency distributions is 2.686, 3.517, 5.893 and 4.938, respectively.

Table A4: The predictive performance of IFL and Dec-IFL when the type embedding dimension $d$ is set to {1,2,4,8,16,32,64,128,256}, evaluated by negative log likelihood (NLL).

| Datasets | SOflow | | MIMIC | | MOOC | | ICEWS | |
|---|---|---|---|---|---|---|---|---|
| | IFL | Dec-IFL | IFL | Dec-IFL | IFL | Dec-IFL | IFL | Dec-IFL |
| $d$=1 | 288.2 | 219.3 | 20.5 | 6.0 | 273.7 | 181.1 | -77.2 | -253.6 |
| $d$=2 | 260.1 | 219.8 | 12.8 | 6.1 | 230.8 | 181.3 | -125.9 | -252.5 |
| $d$=4 | 238.2 | 219.5 | 7.1 | 5.9 | 192.2 | 181.6 | -172.6 | -252.9 |
| $d$=8 | 228.8 | 219.0 | 6.3 | 6.0 | 189.0 | 182.4 | -199.6 | -254.0 |
| $d$=16 | 225.3 | 218.8 | 6.1 | 6.3 | 186.9 | 182.0 | -208.3 | -254.7 |
| $d$=32 | 225.7 | 219.3 | 6.0 | 5.8 | 185.7 | 181.3 | -215.7 | -253.9 |
| $d$=64 | 226.1 | 219.7 | 6.1 | 6.0 | 185.9 | 180.8 | -214.8 | -253.2 |
| $d$=128 | 225.4 | 220.0 | 6.3 | 6.1 | 186.4 | 182.2 | -214.6 | -252.7 |
| $d$=256 | 226.9 | 219.6 | 6.1 | 6.2 | 187.0 | 181.5 | -215.4 | -253.0 |

Table A5: The predictive performance of THP and Dec-THP when the type embedding dimension $d$ is set to {2,4,8,16,32,64,128,256,512}, evaluated by negative log likelihood (NLL).

| Datasets | SOflow | | MIMIC | | MOOC | | ICEWS | |
|---|---|---|---|---|---|---|---|---|
| | THP | Dec-THP | THP | Dec-THP | THP | Dec-THP | THP | Dec-THP |
| $d$=2 | 291.9 | 240.9 | 26.2 | 8.9 | 310.2 | 201.5 | -56.5 | -181.3 |
| $d$=4 | 264.2 | 231.5 | 14.8 | 6.9 | 254.6 | 190.8 | -115.8 | -201.5 |
| $d$=8 | 250.3 | 225.7 | 9.7 | 7.2 | 236.9 | 187.4 | -156.2 | -209.3 |
| $d$=16 | 241.6 | 226.1 | 7.4 | 7.0 | 223.2 | 189.2 | -177.3 | -207.3 |
| $d$=32 | 237.7 | 226.3 | 7.0 | 6.9 | 214.7 | 188.3 | -184.7 | -207.9 |
| $d$=64 | 236.1 | 225.3 | 6.9 | 7.1 | 208.9 | 187.8 | -187.3 | -208.2 |
| $d$=128 | 235.4 | 224.8 | 6.8 | 7.1 | 204.0 | 187.2 | -189.6 | -208.7 |
| $d$=256 | 236.0 | 225.6 | 7.0 | 7.0 | 202.3 | 188.5 | -190.0 | -208.4 |
| $d$=512 | 235.7 | 226.7 | 6.9 | 6.9 | 203.9 | 187.9 | -189.4 | -209.1 |

of how socio-political events have affected one another. For example, event type "Express intent to provide humanitarian aid" and event type "Express intent to provide military protection" have close vector representations in Fig. A6, meaning they have similar influences (probably positive influences) on the event type "Express intent to cooperate". What's more, event type "Threaten with administrative sanctions" have close vector representation with event type "Refuse to release persons or property", meaning they have similar influences (probably negative influences) on the event type "Express intent to cooperate". The results show that our learned event influences are mostly consistent with human experiences.

# D  ABLATION STUDY

Due to the decoupled dynamics for each event type, each EEHD model in our framework has significantly smaller parameter scales compared with traditional models that are designed for full dynamics. What if traditional models use as many parameters as that used in our individual EEHD model and vice versa? Table A4 and A5 answers this and shows that our decoupled EEHD models requires few parameters to capture the corresponding temporal dynamics while they are not adequate for traditional models.

The standard model uses type-specific decoder and keeps the embedding and encoding layer shared across different event types. As a comparison, the embedding, encoding and decoding layer are all type-specific in our proposed decoupled model, to encode the unique dynamics of each event type and thus mitigate the frequency bias of the history vector. This naturally raises a series of questions: what's the performance if only the encoding layer is type-specific? And what's the performance if only the embedding layer is type-specific? We explore all possible variants and report their performance in Table A6, where we see it performs best to let all three modules be type-specific to capture the unique dynamics of the corresponding event type.

Table A6: The predictive performance of variants of IFL and THP. IFL-Emb and THP-Emb use event specific embedding but common encoder-decoder; IFL-E and THP-E use event specific encoder but common embedding-decoder; IFL-EmbE and THP-EmbE use event specific embedding-encoder but common decoder; IFL-ED and THP-ED use common embedding but event specific encoder-decoder. IFL-EmbD and THP-EmbD use event specific embedding-decoder but common encoder. Using this notation, IFL-D and THP-D refer to the original standard model IFL and THP; IFL-EmbED and THP-EmbED refer to our proposed decoupled model Dec-IFL and Dec-THP.

| Model | SOflow | | | MIMIC | | | MOOC | | | ICEWS | | |
|---|---|---|---|---|---|---|---|---|---|---|---|---|
| | NLL | F1 | MSE | NLL | F1 | MSE | NLL | F1 | MSE | NLL | F1 | MSE |
| IFL-D | 225.3 | 30.7 | 2.25 | 6.0 | 65.9 | 0.28 | 185.7 | 39.2 | 4.80 | -215.7 | 27.9 | 0.67 |
| IFL-Emb | 229.2 | 28.1 | 3.75 | 7.2 | 62.3 | 0.38 | 191.3 | 37.2 | 6.18 | -180.6 | 25.1 | 0.72 |
| IFL-E | 227.3 | 30.5 | 2.64 | 6.4 | 65.5 | 0.30 | 188.3 | 38.9 | 5.42 | -201.6 | 27.1 | 0.75 |
| IFL-EmbE | 226.2 | 30.7 | 2.45 | 6.2 | 65.7 | 0.30 | 186.7 | 39.0 | 4.94 | -207.6 | 27.4 | 0.69 |
| IFL-ED | 220.7 | 31.8 | 2.11 | 6.1 | 65.4 | 0.29 | 182.5 | 39.9 | 4.23 | -245.7 | 28.6 | 0.58 |
| IFL-EmbD | 222.2 | 31.4 | 2.26 | 6.1 | 65.5 | 0.29 | 184.0 | 39.6 | 4.58 | -233.4 | 28.4 | 0.63 |
| IFL-EmbED | 219.3 | 32.2 | 2.05 | 6.0 | 65.6 | 0.27 | 181.1 | 40.5 | 4.12 | -253.6 | 29.1 | 0.51 |
| THP-D | 235.4 | 31.5 | 3.55 | 6.8 | 65.7 | 0.35 | 202.3 | 39.6 | 6.11 | -190.0 | 29.4 | 0.84 |
| THP-Emb | 244.8 | 29.3 | 4.33 | 7.9 | 63.7 | 0.47 | 211.0 | 38.3 | 6.83 | -157.4 | 27.2 | 0.91 |
| THP-E | 238.1 | 30.9 | 3.78 | 7.3 | 65.0 | 0.40 | 204.5 | 39.3 | 6.24 | -180.4 | 28.6 | 0.87 |
| THP-EmbE | 237.6 | 31.0 | 3.65 | 7.0 | 65.2 | 0.36 | 203.6 | 39.5 | 6.18 | -186.6 | 29.0 | 0.87 |
| THP-ED | 229.3 | 32.0 | 3.19 | 6.8 | 65.9 | 0.35 | 192.8 | 41.0 | 5.38 | -206.2 | 30.2 | 0.75 |
| THP-EmbD | 232.3 | 31.8 | 3.44 | 6.9 | 65.9 | 0.38 | 196.8 | 40.4 | 5.53 | -197.2 | 29.7 | 0.77 |
| THP-EmbED | 225.7 | 32.4 | 2.97 | 6.9 | 65.7 | 0.37 | 187.4 | 41.3 | 5.23 | -209.3 | 30.6 | 0.70 |

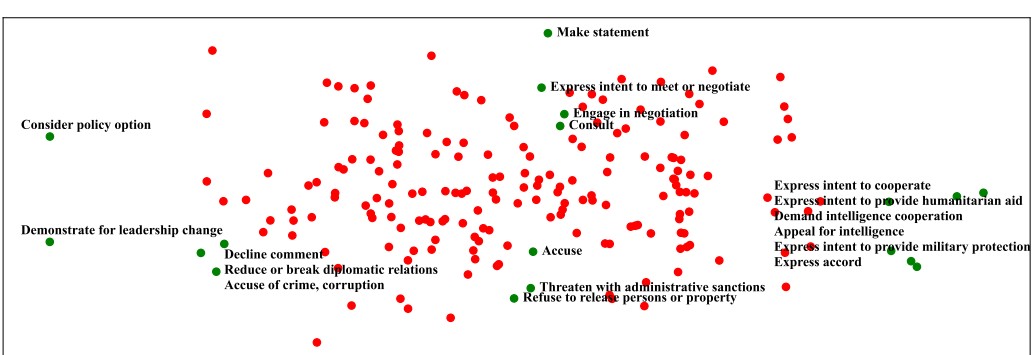

Figure A6: visualization of the embeddings learned by Dec-THP in the vector space of event type "Express intent to cooperate" within the dataset ICEWS.

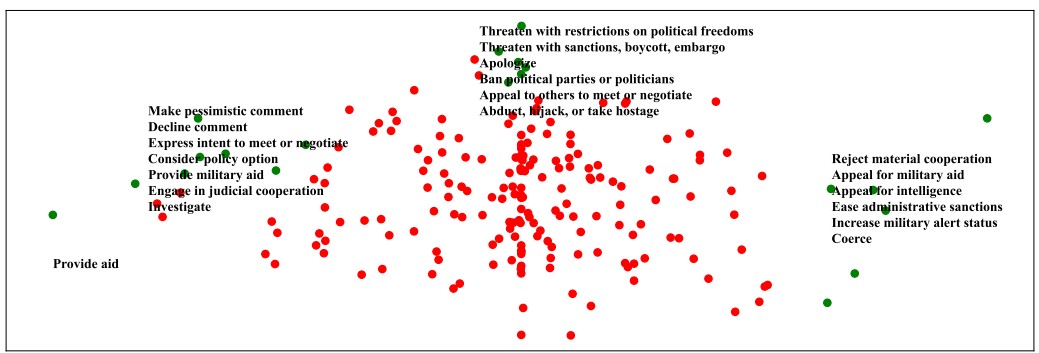

Figure A7: visualization of the embeddings learned by Dec-THP in the vector space of event type "Mobilize or increase armed forces" within the dataset ICEWS.

