# OpenReview forum: "A Decoupled Learning Framework for Neural Marked Temporal Point Process"
_ICLR.cc/2025/Conference — Submitted to ICLR 2025_

### Official Review · Reviewer_iKTe · 2024-11-01

**Soundness:** 2
**Presentation:** 2
**Contribution:** 2
**Rating:** 5
**Confidence:** 4

**Summary:**

The paper introduces a neural modeling approach that independently handles event-specific intensity functions by decoupling the training of event-specific encoder-decoder architectures. This method is designed for time series data, focusing on event times and types, and allows for parallel training. Experimental results on four real-world datasets demonstrate that applying this approach to two previously proposed neural marked temporal process models improves their performance. These models originally relied on a joint intensity function with shared encoder-decoder architectures. The proposed decoupling approach enhances the accuracy of event time and type predictions for these models, speeds up training, and reduces the number of parameters needed.

**Strengths:**

- The paper focuses on the important and impactful problem of modeling event time/type prediction given irregularly sampled time-series data.
- The paper is relatively well-written and easy to follow.
- The proposed decoupling approach seems easy to implement and yields performance improvements over two previously proposed neural marked temporal point process models in terms of event type/time prediction, training speed, and reduced model parameters.

**Weaknesses:**

- The proposed approach seems quite limited in novelty, as it essentially boils down to decoupling the encoder-decoder of previously proposed neural marked temporal process modeling approaches.
- The paper's central claim that models focusing on joint-likelihood estimation of marked temporal point process data fail to account for rare event times compared to the proposed decoupled independent likelihood estimation does not seem substantiated, both in terms of theory and experimental results. Experimental results show general performance improvements (Table 2), but predictions for less frequent events do not seem to improve in general (Table 3). Furthermore, it is unclear if the proposed approach results in statistically significant improvements, as no error bars are provided and only two previously proposed methods are considered for decoupling.
- The proposed independent likelihood approach makes a strong conditional independence assumption, which could be violated in practice; that is, given the past event times and types and the parameters of each specific intensity function, the occurrence of an event is assumed to be independent of other events. This approach may fail to capture the complex dependencies and interactions between different event types that a joint likelihood model could naturally represent.
- The paper does not benchmark against approaches that explicitly model event-specific triggering kernels (unlike RNN or transformer-based approaches) within a joint likelihood, including [1, 2]. Without such comparisons, it is difficult to ascertain the significance of this work.

**Minor**
- Table 1: Add definitions of the hyperparameters to the caption. What is $d_h$?
- Lines 281–292: Make $z$ bold symbol since these are vectors, for consistency.
- Lines 122–133: Define $m$.


**References**
-  [1] Yamac et al. (2023), "Hawkes Process with Flexible Triggering Kernels", MLHC.
-  [2] Pan et al. (2021),  "Self-adaptable point processes with nonparametric time decays", NeurIPS.

**Questions:**

- Table 4: Could you clarify why decoupled encoder-decoder models result in fewer parameters compared to their counterparts? Could you provide the experimental results when the encoder-decoder parameters of the decoupled networks are equivalent to those of the shared networks?
- Figure 3: It's difficult to compare the provided matrices. Given that the data is generated from a known Hawkes process, could you provide plots for the predicted triggering kernels $\phi_k(k_i, t_i)$ versus the ground truth for all the models?
- Table 2: Could you provide results of the decoupling approach applied to all the baselines instead of just to IFL and THP, along with the corresponding error bars?
- Could you benchmark against alternative flexible approaches for modeling triggering kernels, including [1, 2]?
- Theorem 2: I don't think the joint likelihood approach is equivalent to the independent likelihood unless the conditional independence assumption holds; i.e., given the past event times and types and the parameters of each specific intensity function, the occurrence of an event is independent of other events. Could you clarify the equivalence of the decoupled versus shared learning approaches? Also, if the theoretical equivalence holds, why do the empirical results indicate improved performance of the decoupling approach?

---

> ### Author Response · Authors · 2024-11-21
>
> We really appreciate all the valuable feedback you've given us. We hope our response below can help sort out your concerns.
>
> Q1: The decoupled model can have fewer parameters because each single model is only responsible for one event type. For historical events, some of them have influences on event type $i$, some of them have influences on event type $j$, some of them have influences on event type $k$, and so forth. In our decoupled framework, we have a complete EEHD model for each event type $i$. And in each dedicated model, the encoder only needs to encode the historical events that have influences on event type $i$ and ignore other historical events. While for the standard model, all dynamics of historical events are encoded into one single hidden vector. As a result, larger hidden vector, encoder and decoder are required. We have provided the experimental results when the encoder-decoder parameters of the decoupled networks are equivalent to those of the shared networks in Table A4 and A5 of the updated submission.
>
> Q2: Maybe there exists a misunderstanding. In Figure 3, we are showing the clustering effects of our decoupled framework. That is, if two event types have similar influences on event type $i$, then their embedding in the EEHD model of event type $i$ should be close. We have a detailed explanation for the meaning of Figure 3 in our reply to reviewer Aaqp (“Q1”). If we still miss some details, don't hesitate to let us know.
>
> We know you are expecting us to draw plots like Figure 3 in reference [1]. We understand how to plot the kernels for their model, i.e., kernel $q$ in Equation 8 of their paper, but to be honest, we don’t know how the authors plot these kernels for other models like THP (Figure 3 of their paper) as the authors provided inadequate details in Section 7 of their paper. THP (or SAHP, etc.) doesn't model the triggering kernel of pairs of event types, but the model in reference [1] does, i.e., the $q$ in Equation 8 of their paper.  Since THP doesn't model that, neither does our decoupled model Dec-THP. (Our framework only decouple the learning of THP but doesn't alter the architecture of the original model.)
>
> Q3: Beyond IFL and THP, I have provided more results in Table A3 in the appendix of the updated submission.
>
> Q4: Like our method, the two models in reference [1] and [2]  are also generalized Hawkes process and are not standard EEHD models. The difference is that our proposed method is a framework and not limited to a specific encoder and decoder while these two benchmarks both specifically use a kernel followed by a summation as the Encoder (Equation 6 in reference [1] and Equation 4 in reference [2]), like the standard Hawkes process. Actually, we have benchmarked this kind of method (ODETPP in Table 2 of our paper).  Here's the performance of the two models in reference [1] and [2].
>
> ||SOflow    | MIMIC |  MOOC | ICEWS |
> | -------- | -------- | -------- | -------- |-------- |
> | GSHP[1]| 230.7/31.2/2.91|6.3/65.2/0.33|190.8/39.5/4.56|-198.4/28.4/0.69|
> | SPRITE[2]| 232.3/30.9/3.34|6.6/64.3/0.33|194.9/38.2/5.01|-193.6/27.9/0.73|
>
> GSHP is better than SPRITE  as SPRITE captures only time decaying effects while GSHP further captures local effects (see Figure 2 in reference [1]). Both of them underperform our model because of their less expressive encoder. (To obtain the total influence of historical events, they simply add up the influence of each historical event like the standard Hawkes process.)
>
> Q5: Maybe there exists a misunderstanding. Actually, we have no assumptions. In our framework, the occurrence of an event is dependent rather than independent of other events. As shown in Figure 2, when predicting event type $a$, we will evaluate which kind of historical events have influences on event type $a$, i.e., events painted with yellow color $(k_2=2,t_2)$ and $(k_4=K,t_4)$, and use a tailored encoder to encode them into the hidden vector. Finally, we use a tailored decoder to decode the future dynamism of event type $a$. For the discussion regarding Theorem 2, kindly refer to our reply to reviewer Aaqp (“W”).
>
> [1] Yamac et al. (2023), "Hawkes Process with Flexible Triggering Kernels", MLHC.
>
> [2] Pan et al. (2021), "Self-adaptable point processes with nonparametric time decays", NeurIPS.

---

> > ### Comment · Reviewer_iKTe · 2024-11-26
> > **Response by Reviewer**
> >
> > Thank you for the rebuttal and for providing additional results. However, most of my concerns have not been addressed, namely:
> >
> > - The proposed approach seems quite limited in novelty, as it essentially boils down to decoupling the encoder-decoder of previously proposed neural marked temporal process modeling approaches.
> > - The paper's central claim that models focusing on joint-likelihood estimation of marked temporal point process data fail to account for rare event times compared to the proposed decoupled independent likelihood estimation does not seem substantiated, both in terms of theory and experimental results. Although experimental results show general performance improvements (Table 2), predictions for less frequent events do not seem to improve overall (Table 3).
> >
> > For these reasons, I am maintaining my score.

---

> > > ### Author Response · Authors · 2024-11-27
> > >
> > > Thank you for your reply, we hope the further clarification can help.
> > >
> > > (Regarding the novelty) As you have recognized, we proposed a decoupled framework, which  decouples the learning of standard EEHD models by decoupling the embedding-encoder-history vector-decoder. We think our work is novel for two reasons.  First, we observe that the training loss is dominated by the common event types while the loss from other event types may not have been converged (Figure 4). (So we use asynchronous (decoupled) training (learning) for each event type.) Second and interestingly, the embedding in the EEHD model of event type $k$ can reflect the influences of other event types on event type $k$. We illustrate this in Figure 3, Figure A6 and Figure A7. This is very like “influence2vec”, which we believe will have profound impacts on data mining and knowledge discovery.
> > >
> > > (Regarding the performance) We show in Theorem 2 that the expressive power of the decoupled model is equivalent to that of the standard model. But in practice, we show that the training loss is dominated by the common event types while the loss from other event types may not have been converged (Figure 4). In Table 3, Dec-IFL outperforms IFL for 10 event types on dataset SOflow, whereas IFL surpasses Dec-IFL for only 4 event types; Dec-THP outperforms THP for 12 event types on dataset SOflow, whereas THP surpasses Dec-THP for only 5 event types. As we can see, our method can “invoke” more event types than the standard model, the reviewer please kindly refer to Figure A1, Figure A2, Figure A3, and Figure A4 in the appendix for the comparison of type-specific prediction results on datasets MOOC (97 event types) and ICEWS (201 event types). One may doubt that why our decoupled model doesn’t outperform the standard model for every event type. This is because when predicting the next event, all intensity functions are racing to see which has the maximum intensity at a specific time. The event type that has the maximum intensity will be predicted. The standard model outputs large intensity functions for several event types that are easy for prediction (usually but not necessarily, these frequent event types). For these event types, our decoupled model may underperform because our model doesn’t have this bias. (We treat different event types equally (regardless of their frequency) with different EEHD models.) The imbalance problem for event types is very serious in real-world datasets (Figure A5); as far as we know, we are the first to make efforts on this issue.

---

### Official Review · Reviewer_CSLN · 2024-11-04

**Soundness:** 2
**Presentation:** 3
**Contribution:** 2
**Rating:** 5
**Confidence:** 4

**Summary:**

This paper proposed a novel decoupled learning framework for neural marked temporal point processes that effectively mitigates the issue of frequency bias.  Each of the types has a vector representation in each latent space. These are used to model the influence of one event type on the same or another event type. Each event type has its own encoder and decoder. By modeling each event type separately within a complete EEHD architecture, this approprach also enables asynchronous parallel training, improving the training speed, but also allows the embeddings to capture the intricate dependencies between event types.

**Strengths:**

(1) The proposed framework disentangles traditional monolithic modeling with event-type-specific individual modeling, which mitigates the issue of frequency bias while also providing more interpretability than neural TPPs.

(2) The experiment results demonstrate a significant enhancement in training speed and better interpretability on both real-world and synthetic datasets.

(3) The authors show that the proposed framework is general enough by proving that the proposed decoupled learning framework is theoretically equivalent to the standard learning framework.

**Weaknesses:**

(1) In the proof of the equivalence of the proposed decoupled learning framework and the standard learning framework, the statement could be more clear: when proving the proposed framework is no weaker in expressive power than the standard learning framework, verify "let NNk in Equation 6 equal to that in Equation 5" more specifically; when proving the standard learning framework is no weaker in expressive power than the proposed framework, verify "let NNk in Equation 5 equal to that in Equation 6" more specifically.

(2) Based on the Theorem 2 of the paper, the expressive power of the proposed framework is equivalent to the standard learning framework, then it would be equally useful when using these two frameworks with enough parameters and appropriate training in principle. The authors should have a clearer discussion about which model is more advantageous to use in which specific situations. One of the motivations of the proposed framework is to mitigate the issue of frequency bias, it would be better to also compare the performance of this issue within these two frameworks under different settings.

(3) When adopting the decoupled learning framework, the required effort for hyperparameter selection is nearly k-fold as for the standard learning framework, especially for highly unbalanced type case and personalized EEHD architecture tailored to each type. Though the hyperparameter can be set equally, the effects of this issue on the model performance should be discussed carefully and also verified by experiments.

**Questions:**

See the Weaknesses.

---

> ### Author Response · Authors · 2024-11-21
>
> Thank you for your valuable feedback, we are hopeful that our upcoming response will effectively tackle your concerns.
>
> Q1: Thank you for your suggestion. When proving the proposed framework is no weaker in expressive power than the standard learning framework, our idea is to construct a decoupled model that can produce the same output with the given standard model. That’s why we said “let $NN_k$ in Equation 6 equal to that in Equation 5”, where $NN_k$ in Equation 6 is the decoder of our framework and $NN_k$ in Equation 5 is the decoder of the standard framework. ($ NN_k$ is the decoder to decode the future dynamics of event type $k$.)
>
> Q2: For the first question in Q2, the reviewer please refer to our reply to reviewer Aaqp (“W”). For the second question about “which model is more advantageous to use in which specific situations”, we have additionally conducted ablation study to see if it is necessary to let the embedding, encoder and decoder be type-specific like we do in our framework. Table A6 of the updated submission givens positive answers in most datasets. In our framework, each event type is treated by a tailored EEHD model, which can therefore invoke more event types that may be under-represented in the standard model, as shown in Figure A1, A2, A3, and A4. For the third question, we here introduce two commonly used techniques we can think about to mitigate the issue of frequency bias of the standard framework.
>
> One method is to normalize the loss. Specifically, we first count the frequency of each event type in the training data, denoted as $\alpha_i$, where $i$ is a specific event type and $\sum_{i=1}^{K} \alpha_i=1$. Then we multiply their reciprocals with the training objective, i.e.,
> $\log \mathcal{L}(\mathrm{S})=\sum_{l=1}^{L} \log \frac{1}{\alpha_{k_l}} \lambda_{k_l}(t_l) - \int_{0}^{T} \sum_{k=1}^{K} \frac{1}{\alpha_{k}} \lambda_{k}(t) dt$. We retrain the model using this objective and report the performance below. (the number in the bracket is the entropy of the frequency distribution of event types and the frequency distribution is reported in Figure A5 of the updated submission.)
>
> ||SOflow(2.7)    | MIMIC(3.5) |  MOOC(5.9) | ICEWS(4.9) |
> | -------- | -------- | -------- | -------- |-------- |
> | Norm-IFL | 226.6/30.2/2.38|6.1/65.7/0.30|183.8/40.2/4.46|-201.4/27.7/0.88|
> | Norm-THP| 237.3/30.9/3.64|7.0/65.3/0.38|192.9/40.7/5.51|-186.6/29.6/0.93|
>
> Another method is to directly normalize the outputs. We don’t retrain the model and directly do the normalization during the inference stage based on IFL and THP. That is, we consider the learned intensity function $\lambda_i(t)$ as $\frac{1}{\alpha_{i}}\lambda_i(t)$. The results of this variant are summarized below.
>
> ||SOflow    | MIMIC |  MOOC | ICEWS |
> | -------- | -------- | -------- | -------- |-------- |
> | Norm-IFL | 234.6/29.8/3.85|7.3/63.2/0.48|186.8/40.0/5.80|-151.4/26.7/0.98|
> | Norm-THP| 243.3/30.1/4.43|7.8/64.0/0.61|234.2/39.8/6.41|-132.6/28.2/0.97|
>
> We see the two normalization methods both work not very well. For the first method, the main reason is that there always exists several event types that dominate the training loss and the training of other event types is actually not converged, as shown in Figure 4. For the second method that needn't retrain, the reason for the failure mainly comes from the propagation of errors: the normalization factor may don’t fit in the learned intensity function.
>
> Q3: We have ever considered this issue when preparing this submission, though we still set the hyperparameters equally in our work (Line 352). As you said, it's actually for simplicity. Otherwise, we have to list a lot of hyperparameter settings in our paper (ICEWS has 201 event types). Setting tailored hyperparameters could improve performance in principle and the optimal hyperparameters can be automatically searched by program. Intuitively, if there are more event types that have influences on event type $i$, then the size of the EEHD model of event type $i$ should be bigger to capture its dynamics. We mainly search the size of the embedding vector, layers of encoder, size of hidden vector, and the learning rate.  For Dec-IFL, their ranges to seach are  {1,2,3,4,8}, {1,2}, {4,8,16,32}, and {0.0005,0.001,005,0.01}, respectively; For Dec-THP, their ranges to seach are  {4,8,16,32}, {3,4,5,6}, {4,8,16,32}, and {0.0005,0.001,005,0.01}, respectively. The optimal results are reported below.
>
> ||SOflow    | MIMIC |  MOOC | ICEWS |
> | -------- | -------- | -------- | -------- |-------- |
> | Dec-IFL | 218.2/32.5/ 2.00 |6.0/ 65.8/ 0.27  |180.3 /40.8 /4.02  |-260.9 /29.4/ 0.50 |
> | Dec-THP| 223.7/ 32.8 /2.62 |6.6 /65.9 /0.37 | 184.4 /41.7 /4.71  |-232.3 /30.9 /0.64|
>
> Almost surely, employing tailored hyperparameters can make the performance better as using unified hyperparameters is just a special case.

---

> > ### Comment · Reviewer_CSLN · 2024-11-27
> >
> > Thanks for the rebuttal and for providing additional experiment results, which are helpful and make the claims of this paper clearer. However, the novelty of the proposed approach (decoupling in parameter sharing over the existing method), the increased complexity for computation, and the derivative inefficiency for the case of rare events (Table 3) are still limited. Therefore, I will maintain my score.

---

> > > ### Author Response · Authors · 2024-11-27
> > >
> > > Thank you for your reply, we hope our further clarification on the novelty and performance can help sort out your concerns.
> > >
> > > (Regarding the novelty) As you have recognized, we proposed a decoupled framework, which  decouples the learning of standard EEHD models by decoupling the embedding-encoder-history vector-decoder. We think our work is novel for two reasons.  First, we observe that the training loss is dominated by the common event types while the loss from other event types may not have been converged (Figure 4). (So we use asynchronous (decoupled) training (learning) for each event type.) Second and interestingly, the embedding in the EEHD model of event type $k$ can reflect the influences of other event types on event type $k$. We illustrate this in Figure 3, Figure A6 and Figure A7. This is very like “influence2vec”, which we believe will have profound impacts on data mining and knowledge discovery.
> > >
> > > (Regarding the performance) We show in Theorem 2 that the expressive power of the decoupled model is equivalent to that of the standard model. But in practice, we show that the training loss is dominated by the common event types while the loss from other event types may not have been converged (Figure 4). In Table 3, Dec-IFL outperforms IFL for 10 event types on dataset SOflow, whereas IFL surpasses Dec-IFL for only 4 event types; Dec-THP outperforms THP for 12 event types on dataset SOflow, whereas THP surpasses Dec-THP for only 5 event types. As we can see, our method can “invoke” more event types than the standard model, the reviewer please kindly refer to Figure A1, Figure A2, Figure A3, and Figure A4 in the appendix for the comparison of type-specific prediction results on datasets MOOC (97 event types) and ICEWS (201 event types). One may doubt that why our decoupled model doesn’t outperform the standard model for every event type. This is because when predicting the next event, all intensity functions are racing to see which has the maximum intensity at a specific time. The event type that has the maximum intensity will be predicted. The standard model outputs large intensity functions for several event types that are easy for prediction (usually but not necessarily, these frequent event types). For these event types, our decoupled model may underperform because our model doesn’t have this bias. (We treat different event types equally (regardless of their frequency) with different EEHD models.) The imbalance problem for event types is very serious in real-world datasets (Figure A5); as far as we know, we are the first to make efforts on this issue.

---

### Official Review · Reviewer_2wWq · 2024-11-04

**Soundness:** 2
**Presentation:** 3
**Contribution:** 2
**Rating:** 3
**Confidence:** 4

**Summary:**

The paper proposes a decoupled learning framework for neural marked temporal point process, where separate EEHD architecture is used for each event type, including learning multiple  event embeddings corresponding to modeling different event types. The proposed approach is reported to have improvements in terms of performance, training time and interpretability.

**Strengths:**

1. The paper proposes to address the problem of applying the standard neural marked temporal point process to real-world data sets where event types are unbalanced. This seems to be an important and novel problem to address.
2. The paper is well written and easy to follow, though some modification in presentation would improve the readability. The experimental results are neatly presented. The Figures clearly demonstrate the proposed approach and the ingenuity of the result presentation (for instance ICLR in Figure 3) is appreciable.

**Weaknesses:**

1. The fundamental idea of decoupling and learning independently across the event types itself seems to be a major weakness.
2. The approach results in linear growth in number of EEHD architecture parameters with respect to event types. This may not be scalable.
3. The approach itself can be a drawback in addressing the performance issue in event types with limited data. Due to decoupling, as separate model parameters are used for each event type,  the proposed approach may not be able to learn well from an event type with limited data. The standard neural marked point process architecture may not suffer much from this as it shares parameters across event types, consequently allowing knowledge across the event types.
4. The approach might be useful for a specific application or data set, but proposing it as a general approach to solve an unbalanced data set case might be a bit far-fetched.
5. The experimental results do not convincingly demonstrate the improved performance of the proposed approach, and perform poorly in some data sets and cases.
6. Not having ablation studies with some variants of the proposed model (event specific embedding but common encoder-decoder, common embedding but event specific encoder-decoder etc. ) makes it difficult to understand what aspect of the decoupling helped in improving the performance.
7. Experimental results does not follow a consistent pattern. The results with baslines are better at mimic 2, and in several places in Table 3, the proposed method didnt bring any improvements on event types with small number of events. The same is teh case with Figure 3 comparing learnt influence matrix with ground truth. There are mismatches in expected outcomes at several places.

**Questions:**

1. Discuss applications where the event types are unablanced, and provide motivation  from the application perspective
2. Any reference to support the statement that standard EEHD methods fail in the data with unbalanced event type
3. How is the performance when variants of the proposed model are used, event specific embedding but common encoder-decoder, common embedding but event specific encoder-decoder, common embedding and encoder but event specific decoder  etc.
4. Kindly provide sampling procedure as an  algorithm, to make it more comprehensible.
5. In the discussion section, connection between standard Hawkes process and the proposed approach is established. But note that, standard Hawkess process has  global shared components while the proposed architecture has no shared components. Sharing of components helps in knowledge transfer and leads to an improved performance.
6. Kindly provide more details on data set creation, is the 60/20/20 split followed per event or across all events, and how exactly each split is obtained from the event sequence.
7. Why is small sequence length of MIMIC 2 a problem specifically for the proposed method ?
8. For experimental comparison, a baseline considering only events of the same type can be considered to see if the proposed method is able to capture the inter-event influences and improve the predictions.
9. Kindly provide an explanation (or equation) detailing computation of values in the learnt influence matrix.
10. Kindly report in Table 4, the total parameter counts of DEC-IFL and DEC-THP across all the event types.
11. On paper presentation, I think it would be better to provide a separate background section discussing previous works and then discuss the proposed methodology.

---

> ### Author Response · Authors · 2024-11-21
>
> We deeply value the feedback you have given, which is extremely useful. We anticipate that our subsequent response will satisfactorily address your concerns and provide the answers you seek.
>
> Q1 and Q2: To see the imbalance of event types and failure of the standard model, the reviewer please refer to Table 3, Figure A1, A2, A3, A4 and A5 (Figure A5 is newly added in the updated submission), where we see some event types occur less than 100 times while some event types occur more than 20000 times, and our method can invoke more event types than the standard model. In Table 3, Dec-IFL outperforms IFL for 10 event types on dataset SOflow, whereas IFL surpasses Dec-IFL for only 4 event types; Dec-THP outperforms THP for 12 event types on dataset SOflow, whereas THP surpasses Dec-THP for only 5 event types.  Similar conclusions can be drawn over datasets MOOC (97 event types) and ICEWS (201 event types) as shown in Figure A1-A4.
>
> Q3: Thank you for your suggestion. Firstly, it should be noted that “common embedding and encoder but event specific decoder” is exactly the standard model. That is, both the standard method and our method use event specific decoder (see $NN_k$ in Equation 5 and 6). Secondly, kindly note that there is no such model that use common embedding, encoder and decoder as it could not output different intensity functions for different event types. Hence, except for the standard model and our model, there are a total of five kinds of variants, as shown in Table A6. The results  show that it performs better to let the encoder  be type-specific to encode the unique dynamics of the corresponding event type and  that it performs worse to let the decoder be shared across event types.
>
> Q5: From our understanding, the standard Hawkes process (Equation 11) has no global shared component. $\alpha_{i,j}$ is dedicated to build the intensity function of event type $i$, encoded by encoder $\phi_i$, which is the same as our case: the local embedding $z_m^i(j)$ is dedicated to build the intensity function of event type $i$, encoded by encoder $Encoder_i$. We are not very understanding your point “global shared” as $\alpha_{i,j}$ is only used in encoder $\phi_i$ to produce $\lambda_i(t)$ and is not used (shared) in other encoders. If we have overlooked anything, we would greatly appreciate it if you could kindly inform us.
>
> Q6: As shown in Line 312 of our paper, each dataset is split into training/validation/testing set according the number of event sequences, with each part accounting for 60%/20%/20%, respectively. Taking dataset SOflow, which has 6633 event sequences (see Table 1), as an example, we randomly select 6633\*0.6=3979 event sequences as the training data, 6633\*0.2=1327 event sequences as the validation data, and 6633\*0.2=1327 event sequences as the testing data.
>
> Q7:The proposed method shows no advantage over the standard model when the sequence length is very short, but this doesn’t mean small sequence length is a problem for our proposed method. (The results of our method and the standard one in this case are similar.) Our method shows no improvement when the sequence length is very short because the hidden vector in standard model can well summarize the very short historical events. Of course, our method can also summarize the historical events well. Another finding on dataset MIMIC, which has 715 sequences in total (Table 1),  is that there are 564 sequences that repeat the same event type. That is, in these sequences, the event type of the next event is always the same as the event type of historical events. In the remaining 151 sequences, there are 128 sequences looking like this (ignoring the event times): (sequence 1) [1, 0, 0, 0], (sequence 2) [13, 1, 1, 1, 1], (sequence 3) [11, 1, 1, 1, 1], etc. As you can see, there are only 2 event types in each sequence and one of them tends to repeat. These kind of simple patterns, together with the too short length to provide enough information about the dynamics, makes our method show no advantage.

---

> ### Author Response · Authors · 2024-11-21
>
> Q9: There is no additional operation to compute the influence matrix, instead, it is simply the local embedding itself. We have provided a detailed introduction to Figure 3 in our reply to reviewer Aaqp ("Q1"). If we still miss some details, don't hesitate to let us know.
>
> Q10:  The total parameter counts of our model is the number of event types multiplies the parameter counts of one single EEHD model. For example, Dec-IFL has 22 $\times$ 1K= 22K parameters over dataset SOflow and Dec-THP has 22 $\times$ 6K = 0.132M parameters. What if traditional models use as many parameters as that used in our individual EEHD model and vice versa? Table A4 and A5 answers this. Please note that each individual EEHD model in our framework can have smaller parameter scales because of decoupled dynamics for each event type. For the EEHD model of event type $k$, it only needs to encode the historical events that have influences on $k$ and ignore others. (For historical events, some of them have influences on event type $i$, some of them have influences on event type $j$, some of them have influences on event type $k$, and so forth.)
>
> Q4&Q8&Q11: Thank you for your suggestion, we will do that for readers' convenience. We are not very understanding Q8, can you kindly provide more details?

---

> ### Comment · Reviewer_2wWq · 2024-11-25
> **Response to Authors**
>
> Thank you for the response and clarifications. The newly added baseline results in  Table A.6 are helpful to get a better picture of the proposed method.  However, the limited technical novelty of the proposed approach (trivial modification of an existing method), the contentious philosophy of complete decoupling over parameter sharing, and the inability to convincingly demonstrate the effectiveness on events with low frequency (Table 3) are still some drawbacks associated with the paper.

---

> > ### Author Response · Authors · 2024-11-26
> >
> > Thank you for your reply. We decouple the learning of the standard method mainly for two reasons. First, we observe that the training loss is dominated by the common event types while the loss from other event types may not have been converged (Figure 4). (So we use asynchronous (decoupled) training for each event type.) Second and interestingly, the embedding in the EEHD model of event type $k$ can reflect the influences of other event types on event type $k$. We illustrate this in Figure 3, Figure A6 and Figure A7. This is very like “influence2vec”, which we believe will have profound impacts on data mining and knowledge discovery. In Table 3, Dec-IFL outperforms IFL for 10 event types on dataset SOflow (22 event types), whereas IFL surpasses Dec-IFL for only 4 event types; Dec-THP outperforms THP for 12 event types on dataset SOflow, whereas THP surpasses Dec-THP for only 5 event types. As we can see, our method can “invoke” more event types than the standard model, the reviewer please kindly refer to Figure A1, Figure A2, Figure A3, and Figure A4 in the appendix for the comparison of type-specific prediction results on datasets MOOC (97 event types) and ICEWS (201 event types). The imbalance problem is very serious in real-world datasets (Figure A5); as far as we know, we are the first to make efforts on this issue.

---

### Official Review · Reviewer_Aaqp · 2024-11-04

**Soundness:** 2
**Presentation:** 3
**Contribution:** 2
**Rating:** 6
**Confidence:** 4

**Summary:**

The authors propose to model event types in a marked point process individually to alleviate issues associated with (heavy) event sitribution imbalance. Specifically, each event is modeled via a embedding-encoder-history vector-decoder (EEHD), thus models for each event type can be learned in parallel. Experiments on real-world and synthetic datasets demonstrate that the proposed approach achieves state-of-the-art performance in prediction tasks while significantly increasing training speed by a factor of 12 relative to the monolitic EEHD architecture.

**Strengths:**

The proposed approach has two advantages: i) it is very simple by essentially treating the modeling of intensity functions completely separately; and ii) requiring model specification with less parameters thus making the training process faster.

**Weaknesses:**

The authors show in Theorem 2 that the proposed and standard learning frameworks are equivalent, which then implies that there is not a reason for the proposed model to perform better (in general) than the standard model. One may argue that the proposed model is easier to train, but the theorem basically guarantees that one can always find a parameterization of the standard model that matches that of the proposed model and vice versa.

The experiments presented by the authors have several issues:
- The choice of the hyperparameters for the models compared in Table 2 is not clear, especially for IFL, Dec-IFL, THP and Dec-THP. For instance, embedding sizes are too different between the standard and the proposed approach, and though there are ablation results in Tables A3 and A4, they consider a very limited range for d. Also, there is no language about learning parameters and regularization strategies.
- The results in Table 4 also need additional explanation. For instance, the text reads that for the proposed approach training speed is "the lowest for K decoupled models", however, how were models trained, serially or in parallel, and how it is guaranteed that training speeds across models are comparable?
- The experiments in Table 2 (and presumably Table 3) are averages over 10 runs but their variation is not reported, which will be important to understand the significance of the performance differences.
- The authors emphasize that the proposed model is advantageous in situation where event incidence is imbalanced, however comparisons with approaches that address imbalance (some of which are mentioned in the related work) are not considered.

**Questions:**

It will be useful to further explain Figure 3 because as is, the point the authors are trying to make is not very clear. Also, how do these compare with the standard model?

How is the NLL calculated in the experiments? Further, one wonders why NLLs do not seem to be consistent in Table 2, Figure 4 and Table A3?

---

> ### Author Response · Authors · 2024-11-21
>
> We greatly appreciate the valuable feedback you have provided. We hope that our following response will adequately address your concerns and answer your questions.
>
> W: As you said, the theorem basically guarantees that one can always find a parameterization of the standard model that matches that of the proposed model and vice versa. In the proof, we see that the proposed model only needs to copy the embedding, encoder and decoder to match the standard model, which is very easy to realize. Conversely, it requires the universal approximation theory for the standard model to produce the same outputs as that of our proposed model. In this process, the standard model should carefully encode and decode all historical events as any of them could be of significance for one specific event type. Practically, however, the training tends to favor those most common events as they dominate the loss. Moreover, by joint training, we see the component loss of the standard model usually cannot be converged, as shown in Figure 4. RNNs vs. Transformers present a similar case to us. Although theoretically, there exists an RNN that can achieve the same performance as a given Transformer (both of them are universal approximators for sequence), it is always hard for the RNN to find the desired parameters in practice.
>
> W1: We utilize a very small embedding dimension as well as a history vector dimension because each individual model in our framework only needs to process one event type. Doing so, it accelerates the training speed (Table 4), which is one of the contributions of our study. Of course, if you prefer, you can set a very large embedding dimension, history vector dimension, encoder, and decoder. However, this may not improve the model's performance because the performance has already been saturated, as demonstrated in Tables A4 and A5 of the updated submission. In these two tables, we see the decoupled models work well with small dimensions and increasing $d$ does not provide significant gains. We use the Adam optimizer to update the model parameters, with the weight_decay set to 1e-5 and the learning rate set to 0.001.
>
> W2: As mentioned in Section 2.4 of our paper, the $K$ decoupled models are trained in parallel. Specifically, we have 4 machines equipped with 8 GPUs (24GB) and 80 CPU cores. And for the four real-world and four synthetic datasets we used in our work, one machine is already enough as it occupies very little GPU memory for one single EEHD model (Table 4). For each EEHD model, we calculate the average training time per epoch, i.e., the time elapsed from the first epoch to the last epoch divided by the number of epochs. We report the slowest training time per epoch among the $K$ individual EEHD models as the training time (per epoch) of the entire model.
>
> W3&W4: We have provided the error bars in Table A3 of the updated version of our submission. In Line 466 of our paper, we stated that to our knowledge, we are the first to address the issue of frequency bias. If we have overlooked anything, we would greatly appreciate it if you could kindly inform us.

---

> ### Author Response · Authors · 2024-11-21
>
> We here answer the questions you raised.
>
> Q1: To clarify this, let’s start with some important background (they can also be found in Line 110 of our paper). In standard neural MTPPs, the embedding layer assigns each event type a vector representation and the learned representations can naturally group similar event types according to the spirit of Word2vec. However, this kind of  embedding can not reflect the dependencies between different event types. For example, which event types have similar influences on the given event type $k$? Their embedding layer has size $K\times d$, where $K$ is the number of event types and $d$ is the embedding dimension.
>
> As a comparison, we create $K$ vector spaces and in each, an event type will have a vector representation, which we call local embedding. That is, we have $K$ EEHD models and in each model, the embedding layer has size $ K \times d$. If we consider the $K$ EEHD models as a system, then the embedding layer of this system has size $ K \times K \times d$. (Note that our embedding dimension $d$ can be very small because of the decoupled dynamics for each event type, see our response for W1.)  Using our method, an event type can have different vector representations (embeddings) in different EEHD models, indicating that it can have different influences on different event types. In EEHD models of event type $k$, if event type $a$ and event type $b$ have close embeddings, then event type $a$ and event type $b$ have similar influences on event type $k$ according to the spirit of Word2vec. We illustrate an example in Figure 2, where the embedding of event type 2, event type 5, event type $K-2$ and event type $K$ are close (all painted yellow) in the EEHD model of event type $a$, indicating that these four event types have similar influences on event type $a$. But in the EEHD model of event type $b$, it is event type 1, event type 5 and event type $K-1$ that have similar influences on event type $b$.
>
> By now, we can start talking about Figure 3, which illustrates the relationships between different event types, just like Figure 2. Specifically, we conduct experiments on four datasets generated by Hawkes Process (Equation 11), namely Haw1, Haw2, Haw3, and Haw4, where the ground truth influences among event types are known. In Figure 4, the upper four matrices present the configurations of parameters $\alpha_{i,j}$ for the four Hawkes datasets. The parameters $\beta_{i,j}$ are uniformly set to 2.5 across all instances and all datasets. The parameter $\alpha_{i,j}$ quantifies the magnitude of influence that event type $j$ exerts on event type $i$, analogous to the embedding of event type $j$ in the EEHD model of event type $i$. Recall that the embedding dimension for Dec-IFL is set to 1 (see the caption and Table A1), thus the embedding of our whole system has size $K\times K\times d$=$K\times K \times 1$=$K\times K$. We present them by the lower four matrices in Figure 3 (rounded to one decimal place), where each row corresponds to an EEHD model. The element in the $i^{th}$ row and $j^{th}$ is the embedding of event type $j$ in the EEHD model of event type $i$.
>
> Next, we provide an example to analyze the results. For dataset Haw4, according to the ground truth influence parameters $\alpha_{1,2}=1.5$, $ \alpha_{1,3}=1.5$ and $ \alpha_{1,4}=1.5$, event type 2, event type 3, and event type 4 all exert influences on event type 1. This kind of relationship is well reflected by their embeddings in the EEHD model of event type 1. The embedding of event type 2, event type 3 and event type 4 in the EEHD model of event type 1 are -0.4, -0.8, and -0.8, respectively. They have close embeddings! Conversely, event types that have no influence on event type 1 are also clustered together. The embeddings of event type 1 and event type 5 in the EEHD model of event type 1 are 0.2 and 0.3. For more results, the reviewer can kindly refer to Figure A6 and A7 in the appendix, where we illustrate the embeddings learned by Dec-THP on dataset ICEWS.
>
> Q2; NLL is the negative log-likelihood, and the log-likelihood of one sequence is calculated in Equation 7. In testing data, we have multiple sequences and we report the averaged NLL per sequence. In Table 2, Table A4 and Table A5, we report the results on the testing data as we are reporting the predictive performance. (We have updated Table A4 and Table A5 in the updated submission.) In Figure 4, we show the training (convergence) process as the epoch increases, and thus we are reporting the NLL results on the validation data.

---

> > ### Comment · Reviewer_Aaqp · 2024-11-27
> >
> > Many thanks for the detailed response. Though the authors basically confirmed my points about Theorem 2 and Q1, the answers to W1, W3, W4 and Q2 are satisfactory, thus I am modifying my score accordingly.

---

> > > ### Author Response · Authors · 2024-11-28
> > >
> > > Thank you for your reply! We hope our responses above have addressed your concerns and questions. If we have still missed something about Theorem 2 (“W”) and Figure 3 (“Q1”), don’t hesitate to let us know and we are looking forward to further discussion with you.

---

### Meta-Review · Area_Chair_HAk6 · 2024-12-23

**Metareview:**

This paper proposes a decoupled learning framework for neural marked temporal point processes (MTPPs) to address frequency bias in event type modeling. The authors claim their approach mitigates underrepresentation of rare event types, enables asynchronous parallel training, and improves interpretability. The framework independently models each event type using a separate Embedding-Encoder-History vector-Decoder (EEHD) architecture. The authors report improved performance on several datasets compared to standard MTPP models, along with significant training speedups.

The paper addresses the important issue of frequency bias in MTPP modeling and demonstrates faster training through asynchronous parallelization. The authors attempt to improve interpretability via event-specific embeddings and show some performance improvements on multiple datasets.

However, the paper suffers from limited technical novelty, as the approach primarily decouples existing architectures. The benefits of complete decoupling versus parameter sharing are questionable, and the increased complexity in hyperparameter tuning and model selection is a concern. Moreover, the improvements for rare event types are inconsistent across datasets, undermining a key claim of the paper.

The most important reasons for rejecting this paper are its limited technical novelty and contribution to the field, inconsistent improvements for rare event types, and lack of convincing evidence that the proposed approach significantly outperforms existing methods across a wide range of scenarios.

**Additional Comments On Reviewer Discussion:**

During the rebuttal period, reviewers expressed concerns about the limited novelty of the decoupling approach, questionable benefits for rare event types, increased complexity in hyperparameter tuning, and the theoretical equivalence to standard models conflicting with empirical improvements. The lack of comprehensive comparisons with other methods was also noted.

The authors responded by arguing that decoupling allows for tailored modeling of each event type's dynamics and provided additional results showing improvements for some rare event types. They demonstrated results with both equal and tailored hyperparameters across event types and explained that practical improvements arise from training dynamics despite theoretical equivalence. The authors also included comparisons with additional baseline methods.

Despite these efforts, the reviewers maintained their scores, indicating that the paper remains marginally below the acceptance threshold. The Area Chair's comment suggests that the concerns raised by the reviewers have not been adequately addressed, and the paper is likely to be rejected.

---

### Decision · Program_Chairs · 2025-01-22

Reject